# National-scale biogeography and function of river and stream bacterial biofilm communities

**Amy C. Thorpe** [1] ✉, **Susheel Bhanu Busi** [1] ✉, **Jonathan Warren** [2], **Laura H. Hunt** [2], **Kerry Walsh** [2] & **Daniel S. Read** [1] ✉

Biofilm-dwelling microorganisms coat the surfaces of stones in rivers and streams, forming diverse communities that are fundamental to biogeochemical processes and ecosystem functioning. Flowing water (lotic) ecosystems face mounting pressures from changes in land use, chemical pollution, and climate change. Despite their ecological importance, the taxonomic and functional diversity of river biofilms and their responses to environmental change are poorly understood at large spatial scales. We conducted a national-scale assessment of bacterial diversity and function using metagenomic sequencing from rivers and streams across England. We recovered 1,014 metagenome-assembled genomes (MAGs) from 450 biofilms collected across England's extensive river network. Substantial taxonomic novelty was identified, with ~20% of the MAGs representing novel genera. Here we show that biofilm communities, dominated by generalist bacteria, exhibit remarkable functional diversity and metabolic versatility, and likely play a significant role in nutrient cycling with the potential for contaminant transformation. Measured environmental drivers collectively explained an average of 71% of variation in the relative abundance of bacterial MAGs, with geology and land cover contributing most strongly. These findings highlight the importance of river biofilms and establish a foundation for future research on the roles of biofilms in ecosystem health and resilience to environmental change.

It is estimated that up to 80% of all bacterial and archaeal cells on Earth exist within biofilms[1]. These biofilms dominate microbial life in freshwater streams and rivers[2], forming complex, interacting assemblages of taxonomically and functionally diverse heterotrophic and phototrophic microbes. These include bacteria, algae, fungi, protozoa, and viruses[2], existing within an extracellular polymeric matrix that adheres to the surfaces of stones, aquatic plants, and sediments[1]. Biofilms are hotspots of metabolic activity, driving essential biogeochemical processes such as nutrient cycling, primary production, and respiration[3]. They play key roles in degrading pollutants, regulating water quality[4], and supporting energy flow to higher trophic levels[5]. Microbes within

biofilms are therefore critical to the functioning of river ecosystems[6], which are under increasing pressure from pollution and climate change[7]. Monitoring biofilm microbial communities offers valuable opportunities to assess ecosystem health, detect early signs of pollution, and inform restoration strategies, such as water quality monitoring, bioremediation, and improved wastewater treatment[8]. Consequently, a detailed understanding of the diversity and function of biofilm microbial communities in rivers is essential for the effective management and conservation of these vital ecosystems[4].

Diverse microbial communities have been detected in river biofilms[2,6]. Their composition and functionality are likely influenced by

[1]UK Centre for Ecology & Hydrology, Benson Lane, Crowmarsh Gifford, Wallingford, UK. [2]Environment Agency, Horizon House, Bristol, UK.
✉e-mail: amytho@ceh.ac.uk; susbus@ceh.ac.uk; daniel.read@ceh.ac.uk

a complex interplay of factors, including physiochemical parameters such as temperature, pH, and dissolved oxygen[9–11]; the concentrations of nutrients, organic matter, and pollutants[12–14]; flow dynamics[15]; land use[16,17]; ecological processes such as selection and dispersal[18]; and biotic interactions including cooperation and competition between microbes, and grazing by macroinvertebrates[19,20]. By shaping microbial community composition and function, environmental changes can significantly affect biogeochemical fluxes and overall ecosystem health.

Despite their ecological importance, the diversity of microbial communities and their functional roles in river biofilms remains poorly understood. Furthermore, identifying the environmental drivers that influence biofilm community dynamics and assessing their resilience to pressures, such as nutrient pollution, presents ongoing challenges. These challenges include the complex and heterogenous nature of biofilm communities[6], which are shaped not only by a multitude of external environmental factors, but also by the unique microenvironment and the intricate network of microbe-microbe interactions occurring within the biofilm[3,21]. Few studies capture a sufficiently comprehensive suite of environmental covariates to fully resolve these relationships. Moreover, much of the existing research on river biofilm communities has focused on individual rivers or catchments, providing valuable insights into local environmental drivers of microbial diversity and community dynamics[22–24]. However, these studies typically comprise a relatively small number of samples and cover a limited geographic area.

In recent years, national-scale studies have been undertaken to capture the wider diversity of freshwater microbes and explore their biogeographic patterns across a broad range of environments[25–28]. These national-scale datasets demonstrate the value of molecular approaches, including metagenomics, for uncovering both the taxonomic and functional diversity of microbial communities, and highlight community dynamics and adaptations in response to environmental drivers across large spatial scales. Collectively, national-scale datasets can provide a more comprehensive understanding of the ecological dynamics of freshwater microbes and their drivers at a global scale. However, there are few examples of such national-scale studies of river biofilm communities, and, to our knowledge, no national-scale metagenomic assessment of river microbes currently exists in England.

The Environment Agency (EA) established the River Surveillance Network (RSN) as a national initiative to monitor and assess long-term changes in the health of English rivers. The RSN comprises 1600 sites, representing the diversity of England's extensive river network. As part of this initiative, 450 river biofilms were systematically collected over a three-year period from 146 RSN sites spatially distributed across England for metagenomic sequencing (Fig. 1A). This comprehensive dataset spanned a latitudinal gradient of 645 km. It encompassed all major land cover types present in the country, ranging from woodlands and grasslands to arable land and urban areas, and represented a variety of catchment geologies (Supplementary Table 1, Supplementary Data 1). In addition to the diverse catchments, the RSN sites spanned considerable variability in their physicochemical conditions, including water temperature, alkalinity, pH, dissolved oxygen, dissolved organic carbon, orthophosphate, nitrate-N, and nitrite-N (Supplementary Table 2, Supplementary Data 1).

This systematic investigation of the river network, coupled with detailed environmental data, enables a robust assessment of microbial biofilms, comparable to existing large-scale efforts[25,26,28,29], capturing their biogeographic distribution, taxonomic diversity, metabolic and functional potential, and environmental drivers. Specifically, this study addresses three fundamental research questions:

(i)   How are river biofilm bacterial communities structured across England's river network?

(ii)  What metabolic capabilities and functional traits underpin their roles in ecological processes such as biogeochemical cycling?

(iii) How do environmental drivers shape river biofilm bacterial communities at a national-scale?

By addressing these questions, this study offers novel insights into the biogeography of river biofilm bacteria and their critical roles in ecosystem functioning. These findings help provide a foundation for effective monitoring, management, and conservation of river ecosystems.

## Results and discussion

### Diversity and composition of river biofilm microbial communities

River biofilms are cosmopolitan in their distribution and known to host high microbial diversity, supported by continuous nutrient and community inputs from upstream[6]. However, our understanding of freshwater biofilm microbial communities is significantly less advanced than that of microbial communities in the water column[25] or sediments[9]. Although several studies have explored the taxonomic composition of river biofilms at the local scale[10,11,23], the overall abundance, composition, and genomic traits of these communities across large spatial scales remain poorly understood.

Using metagenomic data averaging 66.80 million reads per sample (±28.5 million reads SD), we found that bacterial sequences comprised the majority (85.17%) of all metagenomic reads in the river biofilms. Eukaryotes and Archaea represented 11.56% and 2.64% of the reads, respectively (Supplementary Data 2). Because of their dominance in biofilms, we focused our analysis on bacterial communities. The assembled reads were used to reconstruct a total of 4027 bacterial bins (archaeal bins were not recovered), which were dereplicated into 1014 metagenome-assembled genomes (MAGs) (also known as species-level genome bins). Of these, 329 MAGs (32%) were identified as near-complete, 491 (48%) as medium-quality, and 194 (19%) as low-quality (Fig. 1B) with a mean completeness of 86.70% (±9.3% SD) and a mean contamination of 3.57% (±4.1% SD). Mean genome size was estimated to be 3.66 Mbp (±1.4 Mbp SD) and mean GC content was 52.66% (±11.7% SD) (Supplementary Fig. 1, Supplementary Data 3).

The 820 medium or high-quality bacterial MAGs retained for downstream analysis encompassed a diverse range of taxa, including 20 known phyla, 35 classes, 91 orders, 160 families, 311 genera and 46 species. The 46 MAGs assigned to known species according to GTDB-Tk had an average nucleotide identity (ANI) ranging from 95.0 to 95.4% (mean 95.01 ± 0.06). There was also substantial taxonomic novelty detected, with 169 (20.6%) of the recovered MAGs representing previously uncharacterised genera, and 774 MAGs (94.4%) representing previously uncharacterised species with no suitable GTDB-Tk reference (≤65% ANI across all species within the database; Supplementary Data 3) (Fig. 1C). Stringent assessment using only near-complete MAGs revealed that 306 (93%) were novel species-level genome bins, with no previously identified representatives. Across all river biofilms, Pseudomonadota was the most dominant phylum and comprised almost half of the total community, with a mean relative abundance of 48.49% (±14.8% SD) (Fig. 1D). Other abundant phyla included Cyanobacteriota with a mean relative abundance of 15.68% (±16.9%), Bacteroidota with a mean relative abundance of 14.77% (±9.5%), and Actinomycetota with a mean relative abundance of 6.27% (±5.1%). Other less abundant phyla each comprised less than 5% of the total community on average. This community composition aligns with previous studies on benthic biofilms collected from a variety of river types globally, including the groundwater and rain-fed River Thames (United Kingdom)[30] and urban and rural rivers in China[31], and glacier-fed streams in alpine regions such as the Southern Alps (New Zealand) and the Caucasus (Russia)[32]. These studies also reported the dominance of Pseudomonadota and Bacteroidota, which have been reported to play important roles in the

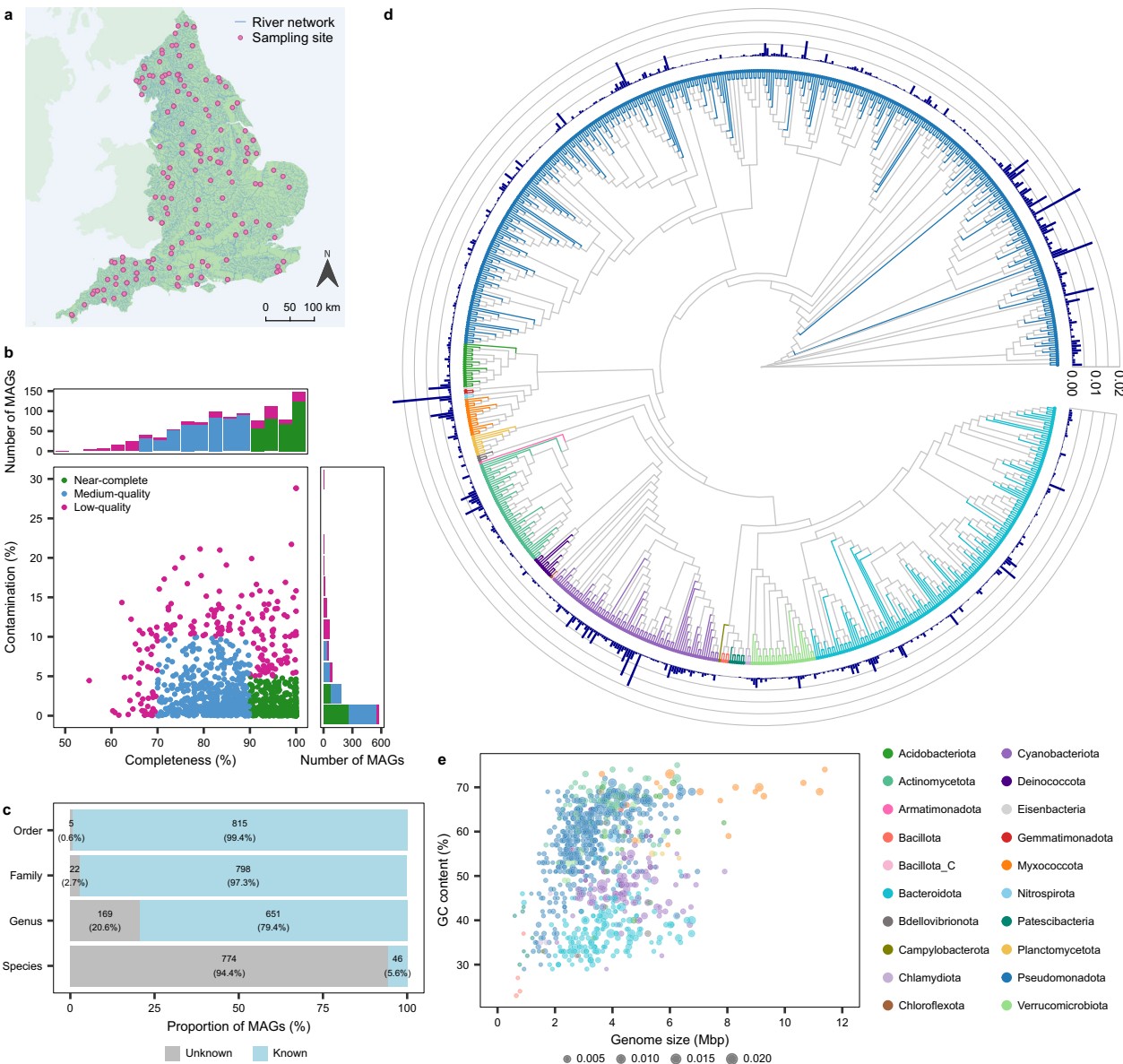

**Fig. 1 | Overview of the diversity, novelty, and genomic characteristics of river biofilm bacterial MAGs. a** Biofilm sampling sites in rivers across England (contains OS data © Crown copyright and database rights 2024). **b** Distribution of MAG completeness and contamination. **c** Taxonomic novelty of MAGs at the order, family, genus, and species level. **d** Phylogenetic tree of MAGs where colours represent different bacterial phyla, and outer bars represent the mean relative abundance of MAGs. **e** Estimated GC content and genome size of MAGs where point size is scaled to mean relative abundance and colour represents bacterial phylum. Source data are provided as a Source Data file.

degradation of organic matter, and Cyanobacteriota, which contributes to primary productivity[33]. In contrast, river surface waters are typically dominated by Pseudomonadota and Bacteroidota, but also by Actinomycetota and Verrucomicrobiota[25,30], highlighting differences in community assembly between benthic biofilms and pelagic habitats.

### Genomic traits of river biofilm bacteria

We found that genome size and GC content were correlated with phylogenetic background, delineating these traits across the bacterial tree of life (Fig. 1E). It was observed that many Myxococcota MAGs are large with a high GC content, Pseudomonadota MAGs are small to medium with a high GC content, and Cyanobacteriota and Bacteroidota MAGs are small to medium with a lower GC content. A similar phylogenetic relationship between GC content and genome size has

been observed in pelagic and sediment bacteria from lakes and rivers[34–36], suggesting that the patterns observed for many phyla such as Myxococcota, Pseudomonadota, Cyanobacteriota, and Bacteroidota are consistent with broader trends across freshwater environments. These traits may reflect ecological niche adaptation. For example, bacteria with larger genomes are typically copiotrophs, with numerous genes associated with complex metabolic processes, allowing them to thrive in fluctuating or nutrient-rich environments[37]. In contrast, those with smaller genomes often have specific nutrient requirements and occupy more stable or oligotrophic environments[37], consistent with genome streamlining theory[38]. Furthermore, lower GC content reduces nitrogen requirements, which may be advantageous for bacteria in nitrogen-limited environments[36]. Genome size is likely affected by a complex interplay of environmental factors, in addition to trophic strategies[39], and biotic interactions[40,41].

Functional annotation of these MAGs revealed multiple adaptations for biofilm matrix colonisation and maintenance, including genes associated with biofilm formation, quorum sensing, chemotaxis, flagellar assembly, the two-component system, exopolysaccharide (EPS) biosynthesis and ATP synthase-binding cassette (ABC) transporters (Supplementary Fig. 2). Notably, many of these genes were enriched in larger MAGs, which is consistent with the functional and metabolic flexibility of copiotrophs and may reflect the greater genomic complexity associated with larger bacterial genomes[36]. These traits are broadly conserved across bacterial biofilms in other freshwater environments, including glacier-fed streams[32] and other cryospheric systems[42], where they have also been found across diverse taxonomic groups, highlighting the phylogenetically widespread contribution to the biofilm matrix through roles in biofilm formation, motility, sensing the environment, and nutrient acquisition.

## Biogeography of river biofilm bacteria

River biofilms host complex communities composed of both generalist and specialist taxa, each fulfilling distinct ecological roles[2]. Generalist taxa are highly adaptable, thriving across a diversity of habitats, and tolerating a broad range of environmental conditions, whereas specialist taxa are adapted to specific habitats, thriving only under specific environmental conditions[43]. Despite the ecological significance of these communities, the biogeographic patterns of river biofilm bacteria are poorly understood and have not yet been addressed in the context of ecological niche space. Using biogeographic mapping and niche breadth analysis coupled with measures of abundance and occupancy, we investigated the spatial distribution and ecological niches of river biofilm bacteria.

The distribution of MAGs across the river network revealed national-scale biogeographic patterns (Fig. 2). Pseudomonadota and Bacteroidota exhibited high relative abundances throughout the country (>5% in more than 90% of sites), dominating biofilm communities. In contrast, Cyanobacteriota displayed regional preferences, with higher relative abundance in the north and southwest of England, and a lower occupancy overall (Fig. 3A), indicating that they occasionally dominate biofilm communities (>5% in only 63% of sites). Other taxa, such as Acidobacteriota, Myxococcota, and Nitrospirota, had a widespread presence but exhibited localised hotspots where their relative abundance was notably higher. Campylobacter and Chloroflexota had more restricted biogeographic distributions and were present at low relative abundances in only 13 and 79 samples, respectively. However, most MAGs were widespread rather than limited to specific locations. Each MAG occupied an average of 413 samples (92%) spatially distributed across the river network, with 176 MAGs (21%) present in all samples. Many of these 176 high-occupancy MAGs were the most abundant members of the community, with a mean relative abundance of up to 2.49% (Fig. 3A). A similar relationship between occupancy and mean relative abundance has been observed in surface water bacterial communities in rivers across North America[25]. However, while river networks comprise mostly high-occupancy taxa, studies of bacterial communities in spatially distributed lakes have found a greater proportion of low-occupancy taxa[26,44]. This difference likely reflects the underlying ecological factors that differ between rivers and lakes, such as habitat connectivity, dispersal dynamics, and environmental stability[45].

Using niche breadth analysis, we further categorised the MAGs as generalists (niche breadth, $B_N$ > median $B_N$) or specialists ($B_N$ < median $B_N$). This metric leverages the relative abundance of MAGs to quantify how evenly they are distributed across the river network, providing a measure of their ecological breadth. Among the 176 high-occupancy MAGs, 88% were identified as generalists. Generalists comprised 58.58% of Pseudomonadota MAGs, and 58.62% of Verrucomicrobiota MAGs, while 51.19% of Cyanobacteriota MAGs, 60.00% of Acidobacteriota MAGs, 64.67% of Bacteroidota MAGs, and 64.71% of

Myxococcota MAGs were categorised as specialists (Fig. 3B). Previous studies have also detected a high abundance of generalist bacteria in river and stream biofilms[2,46,47]. Furthermore, the distance-decay relationship of bacterial community dissimilarity and geographic distance was positive, but weak ($R^2 = 0.04$, $p < 0.001$) (Supplementary Fig. 3A, B), suggesting dispersal rates in bacterial biofilms are high, although other processes could also contribute, such as slow ecological or evolutionary turnover. Nevertheless, some degree of species sorting due to local environmental conditions likely occurs, consistent with the Baas Becking hypothesis, stating that everything is everywhere, but the environment selects[48]. Taken together, these results reveal a dominance of high-occupancy, generalist MAGs across the river network, emphasising the role of rivers as highly interconnected ecosystems with potential for widespread microbial dispersal. The ability of generalist bacteria to tolerate a diverse range of environmental conditions is likely an advantage in lotic ecosystems, where biofilm communities may become dislodged and recolonise further downstream[46].

## Metabolic potential of river biofilm communities

Biofilm communities play critical roles in ecological processes such as primary production, nutrient cycling, and the transformation and breakdown of organic material and pollutants[3,49]. Their contributions to biogeochemical cycling can strongly influence the health and productivity of river ecosystems by mediating the availability of nutrients and other compounds to higher trophic levels in the food web[5]. Previous studies have emphasised the importance of assessing the metabolic capabilities of biofilms to better understand their functional roles in river ecosystems[6]. Despite this, there are few examples of studies that have conducted genome-resolved assessments of microbial metabolic potential within river biofilms[32], particularly across broad spatial scales. As a result, detailed insights into the distribution and prevalence of key metabolic functions, which are necessary to understand the role of biofilm microbes in ecosystem functioning, remain limited.

The majority of bacterial MAGs were found to have the capacity to contribute to carbon, nitrogen, and sulfur cycling in rivers. Among these, Pseudomonadota, Cyanobacteriota, and Bacteroidota MAGs accounted for more than 52.41%, 16.62% and 10.95%, respectively, of the nutrient cycling genes identified in the biofilms (Supplementary Fig. 4). Across the 820 MAGs, the majority of steps in carbon, nitrogen, and sulfur cycling pathways were represented (Fig. 4A–C). Nearly all MAGs encoded for genes involved in organic carbon oxidation (99.39%), and a substantial proportion of the MAGs included genes involved in fermentation (86.83%), acetate oxidation (73.66%), sulfur oxidation (47.93%), and nitrite ammonification (43.29%). Many of these pathways represented by the river biofilms, including organic carbon oxidation and sulfur oxidation, were also prevalent in mountain stream biofilms[50].

Carbon cycling genes, such as *sgdh* (pentose phosphate pathway) and *RuBisCO IV* (carbon fixation), were abundant in members of Pseudomonadota and Cyanobacteriota (Fig. 4D, Supplementary Fig. 5A). Nitrogen cycling genes, including *nirD* and *nirB* (nitrite reductases) (Fig. 4E, Supplementary Fig. 5B), and sulfur cycling genes, including *sdo* (sulfur dioxygenase), *fccB* (sulfur dehydrogenase) and *sat* (sulfate reduction) (Fig. 4F, Supplementary Fig. 5C), were particularly abundant in members of Pseudomonadota, Cyanobacteriota, and Bacteroidota. Additionally, Pseudomonadota and Acidobacteriota MAGs encoded the most diverse range of genes involved in these pathways, with Pseudomonadota encoding 17, 16, and 10 genes and Acidobacteriota encoding 13, 15, and 9 genes involved in the carbon, nitrogen, and sulfur cycles, respectively. The biofilm MAGs also encoded a range of genes involved in oxygen and hydrogen cycling (Supplementary Fig. 5D, E, Supplementary Fig. 6, Supplementary Data 4). Many of these metabolic functions are associated with the production and attenuation of greenhouse gases such as carbon

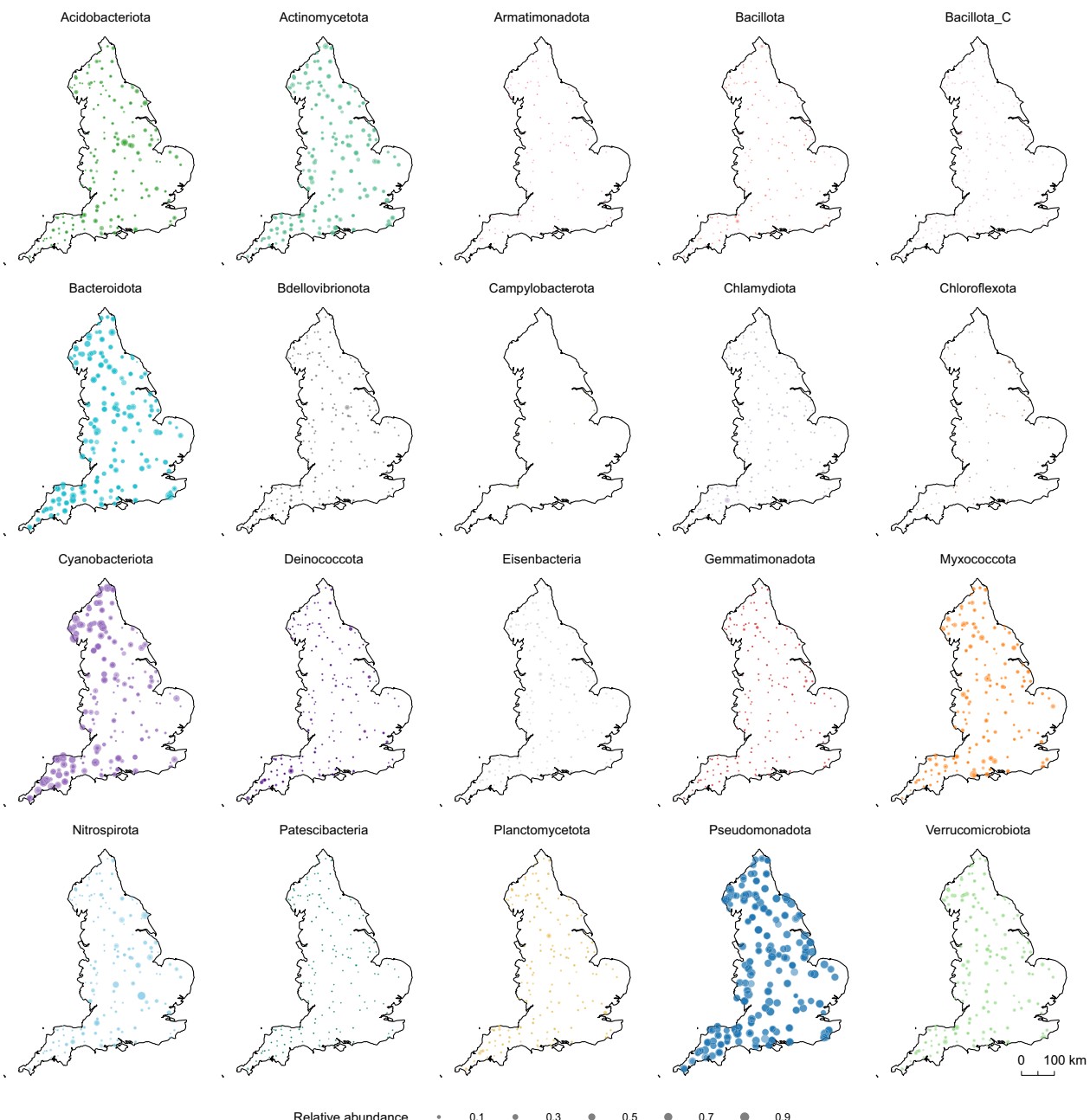

**Fig. 2 | Biogeographic distribution of bacterial MAGs in rivers across England.** Colours represent different bacterial phyla and points are scaled to relative abundance. Source data are provided as a Source Data file.

dioxide, methane, and nitrous oxide[51]. These results highlight the potential of biofilm bacteria to perform diverse biochemical roles, which may include contributing to the flux of greenhouse gases in river ecosystems.

### Functional potential of river biofilm communities

Biofilm bacteria exhibited an extensive range of functional capabilities, including resource acquisition, resource use, and stress tolerance (Fig. 5, Supplementary Fig. 7). Genes for the breakdown of compounds such as simple and complex carbohydrates, cellulose, and proteins were widely distributed across the community (present in more than 74.46% of MAGs) (Fig. 5A). This indicates that the biofilm community can utilise a broad range of organic compounds, highlighting their metabolic flexibility and resilience to fluctuating nutrient availability.

Up to 91.10% of biofilm MAGs contained genes for aerobic respiration, and 21.59% of MAGs, which included members of Pseudomonadota, Actinomycetota, Acidobacteriota, Deinococcota, and Planctomycetota, also possessed genes associated with anaerobic respiration. Additionally, some MAGs, notably those belonging to Pseudomonadota and Cyanobacteriota, had genes for both core and accessory photosynthetic pigments (Fig. 5B). The coexistence of the trophic capabilities identified among biofilm bacteria is consistent with studies showing that biofilms often exist as stratified micro-environments, in which aerobic and photosynthetic bacteria dominate the outer layers exposed to light and oxygen, while anaerobes occupy deeper oxygen-depleted layers[3]. Such trophic diversity may contribute to the versatility of biofilm bacteria, enabling them to occupy multiple ecological niches within the complex biofilm matrix.

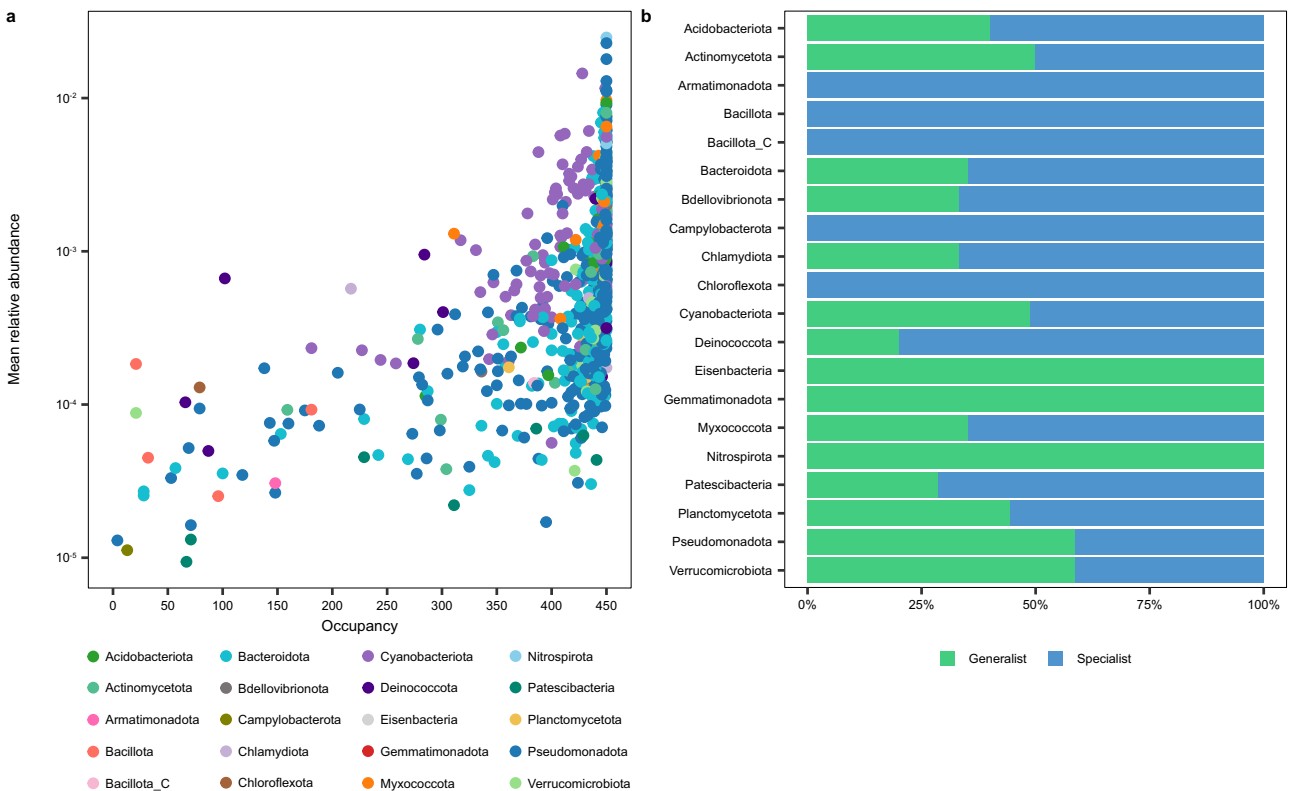

**Fig. 3 | Ecological niche breadth of bacterial MAGs. a** Occupancy (number of samples each MAG is present in) and mean relative abundance (log scale) of MAGs where point colour represents bacterial phylum. **b** Proportion of MAGs of each bacterial phylum classified as generalists (green) and specialists (blue) according to Levins' niche breadth index. Source data are provided as a Source Data file.

Biofilm formation is a key survival strategy for many freshwater microbes, allowing them to gain physical protection from the extracellular polymeric matrix, accumulate and readily access nutrients which may include biopolymers derived from the matrix itself, and interact with other taxa[3]. Further to the observed prevalence of biofilm-associated genes such as those involved in quorum sensing, chemotaxis, flagellar assembly, and EPS biosynthesis (Supplementary Fig. 2), genes associated with biofilm formation were also identified in 96.95% of MAGs encompassing nearly all bacterial phyla detected (Fig. 5C). Exceptions to this were Patescibacteria, which are known for their reduced genomes and symbiotic lifestyle[52] and likely depend on interactions with biofilm-forming bacteria rather than contributing to biofilm formation directly, and Bacillota, which may play a less prominent role in biofilm formation, possibly by relying on an alternative survival strategy such as sporulation[53]. These findings highlight the widespread importance of biofilm formation as a strategy for microbial survival in dynamic river ecosystems, facilitating interactions among taxa, promoting community stability, and acting as a nutrient resource that may buffer potential effects from nutrient fluctuations and other environmental distrubances[3,54]. Additionally, genes involved in a variety of other stress tolerance strategies were identified in the biofilm MAGs, such as compatible solutes accumulation, membrane stability, scavenging of reactive oxygen species, and repair and degradation of damaged proteins (Fig. 5C). Many of these stress tolerance strategies have also been identified in biofilm bacteria in glacier-fed streams[32], highlighting that the widespread presence of stress tolerance genes may support the persistence of biofilm bacteria under fluctuating or unfavourable condition in freshwater environments.

Biofilm communities play an essential role in the uptake, transformation, and degradation of organic compounds, including those that are harmful pollutants in river ecosystems[49]. The vast majority of MAGs (92.20%) contained genes involved in organophosphorus transport (Fig. 5A), which is a common pollutant in rivers derived from agricultural pesticides[55]. A total of 20.49% of MAGs, which included members of Pseudomonadota, Actinomycetota, Deinococcota, Myxococcota, Cyanobacteriota, and Verrucomicrobia, had genes for the transport of aromatic acids. These compounds, while naturally present in ecosystems, can also include polycyclic aromatic hydrocarbons (PAHs) introduced into rivers from industrial activities[56]. Furthermore, 99.39% of MAGs, which encompassed all bacterial phyla detected, had genes involved in the transport of metals. These transportation genes are widespread in bacteria for acquiring essential metals, but they may also participate in the cycling of metals in rivers, which alongside natural sources, may enter rivers through industrial activities[57]. The widespread presence of these genes indicates that biofilm bacteria may be capable of transporting and therefore potentially transforming harmful organic pollutants that pose a significant risk to aquatic life in rivers. Biofilm bacteria may contribute to metal detoxification by immobilising metals and reducing their bioavailability in the water. This demonstrates the importance of biofilm bacteria in maintaining water quality[4] and their valuable application as bioindicators for monitoring ecosystem health[58], particularly as rivers are under increasing threat from pollution and contaminants worldwide[7].

### Environmental drivers of river biofilm communities

Our findings demonstrate the taxonomic diversity of biofilm bacteria and their extensive metabolic and functional roles which are essential for riverine ecosystem functioning. However, while previous studies have explored the environmental drivers of river benthic biofilm communities at the individual catchment scale[10,13,59], large-scale patterns and drivers remain poorly understood[27], particularly in comparison to surface water communities[25]. This knowledge gap limits our

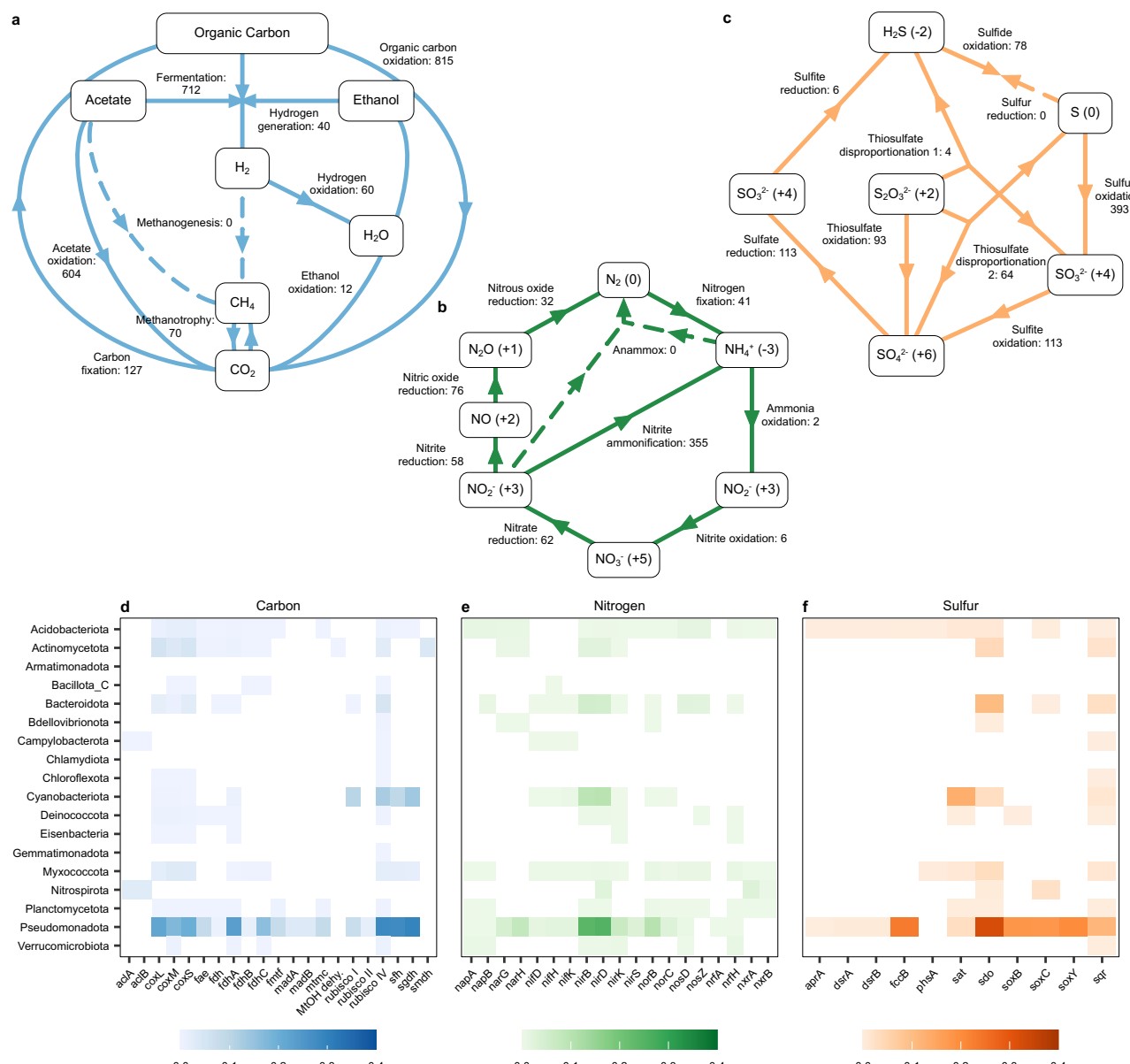

**Fig. 4 | Nutrient cycling and metabolic potential of bacterial MAGs. a** Carbon, **b** nitrogen, and **c** sulfur cycling pathways, identified using METABOLIC, where numbers denote the number of MAGs positive for genes involved in each step and dashed lines show steps with no positive MAGs. Heatmaps of **d** carbon, **e** nitrogen,

and **f** sulfur cycling genes, based on annotations made using metabolisHMM, where colour scales represent gene counts weighted by mean relative abundance and summed by bacterial phylum. Source data are provided as a Source Data file.

ability to understand and predict how biofilms and wider river ecosystems respond to environmental change across diverse landscapes. By incorporating detailed spatial information of the upstream catchment alongside high-resolution monitoring of water chemistry, this study represents a novel and comprehensive effort to assess the environmental drivers of river biofilm bacteria across large spatial scales.

Variance partitioning analysis was used to identify the environmental factors shaping river biofilm community composition across England. Environmental variables were categorised as catchment land cover, catchment geology, water chemistry and watershed characteristics. The total variance in the relative abundance of each MAG explained by the full set of measured environmental variables averaged 70.63% (±19.5%). When grouped by phylum, the total variance explained by all environmental variables ranged from 46.55%

(Bacillota) to 97.36% (Gemmatimonadota) (Fig. 6A). Catchment land cover has previously been shown to significantly influence bacterial community composition in New Zealand stream biofilms[16,27], and in North American river surface water[25]. Consistent with these findings, upstream catchment land cover was also identified as an important driver of river biofilm communities in England, accounting for up to 24.81% of observed variance at the phylum level, with a mean of 12.78% (±7.3%), although variance explained by land cover was particularly low for some phyla such as Gemmatimonadota (0.66%). The most dominant land cover types included acid grassland, improved grassland, and arable and horticulture, although the proportion of variance explained by these factors varied among bacterial phyla (Fig. 6C).

Upstream catchment geology was found to account for the largest proportion of variance in biofilm bacterial community composition, explaining between 10.94 to 93.52% across bacterial phyla, with a mean

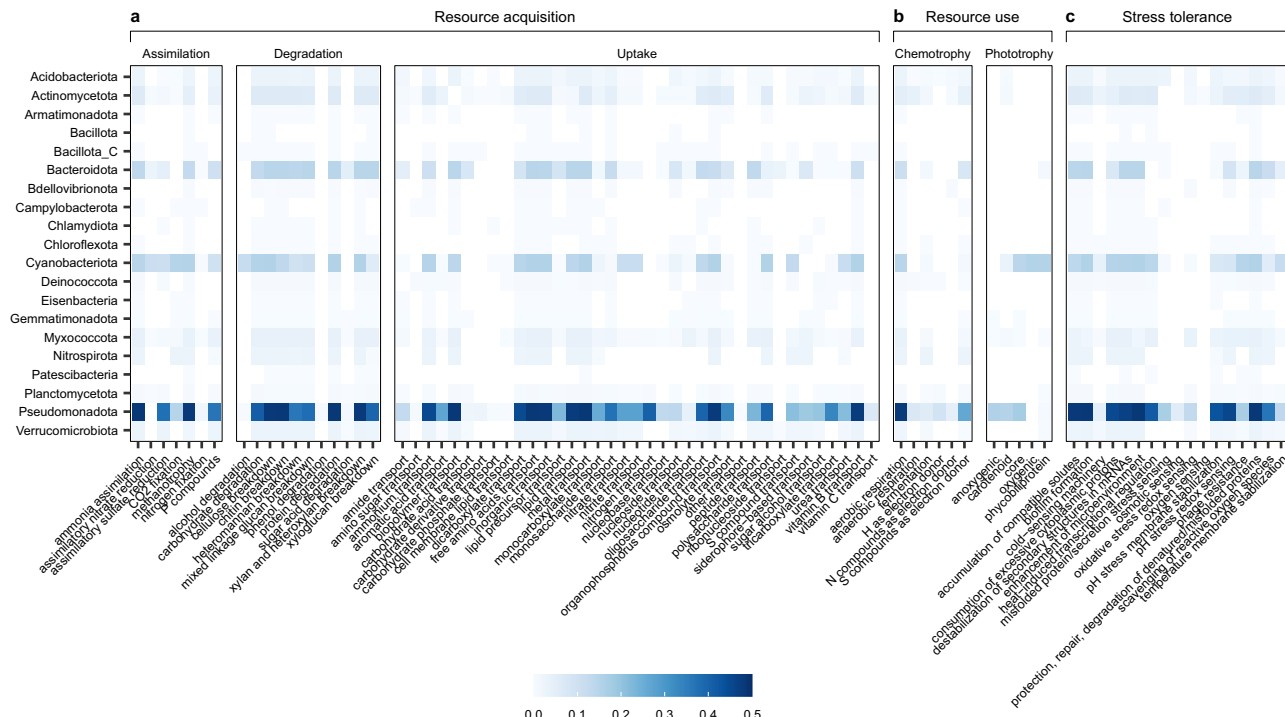

**Fig. 5 | Functional genomic traits of bacterial MAGs.** Traits associated with **a** resource acquisition (substrate assimilation, degradation, and uptake), **b** resource use (chemotrophy and phototrophy), and **c** stress tolerance, based on annotations made using microTrait. The heatmap colour scale represents gene counts per category weighted by mean relative abundance and summed by bacterial phylum. Source data are provided as a Source Data file.

of 45.68% (±18.7%). The proportion of the upstream catchment represented by calcareous and siliceous geologies were identified as the dominant geological drivers, explaining up to 37.46% (mean 18.38% ± 8.4%) and 27.76% (mean 13.55% ± 5.3%) of variation, respectively (Fig. 6D). To further evaluate the influence of these geological types, non-metric multidimensional scaling (NMDS) was performed, which revealed clear clustering of river biofilms along a gradient of calcareous to siliceous geology, suggesting that calcareous and siliceous geology structure bacterial communities in river biofilms differently (Supplementary Fig. 8, Supplementary Data 5). Furthermore, calcareous geology correlated positively with surface water pH ($r = 0.36$, $p < 0.001$), alkalinity ($r = 0.33$, $p < 0.001$), and conductivity ($r = 0.29$, $p < 0.001$), while siliceous geology correlated negatively with these variables ($r = -0.32$, $r = -0.53$, and $r = -0.31$, respectively, $p < 0.001$) (Supplementary Fig. 9, Supplementary Data 5). These findings are consistent with established relationships between geology and river water chemistry, particularly regarding $CaCO_3$-based buffering capacity (i.e., alkalinity), which is higher in catchments underlain by calcareous geology[60,61]. The influence of geology on biofilm communities therefore likely operates through its effects on pH, alkalinity, and conductivity[47,62].

Despite these associations, water chemistry variables explained a relatively small proportion of variance in biofilm community composition (2.41 to 7.45%, with a mean of 4.51% ± 1.6% per phylum). However, individual variables such as alkalinity, conductivity, pH, dissolved oxygen, and water temperature were still influential, each accounting for up to 1.6% of the observed variation (Fig. 6B). While water chemistry was measured and averaged over a three-month period to capture temporal variability, these measurements may still underrepresent the influence of catchment geology, which acts as a persistent landscape-level driver influencing river systems and therefore biofilm development over much longer timescales. Additionally, the biofilm matrix may act as a buffer against physiochemical fluctuations in the water

column[3], reducing the influence of water chemistry on biofilm communities.

In addition to geological and chemical drivers, orthophosphate, nitrate-N, and nitrite-N emerged as key nutrients shaping biofilm community composition, with orthophosphate most notably influencing the relative abundance of Planctomycetota (2.55%) and Bdellovibrionota (2.03%), nitrate-N influencing the relative abundance of Planctomycetota (0.95%) and Nitrospirota (0.90%), and nitrite-N influencing the relative abundance of Bacillota (2.21%) and Bdellovibrionota (1.24%). Wastewater treatment plant (WWTP) load, which correlated positively with orthophosphate ($r = 0.66$, $p < 0.001$), nitrate-N ($r = 0.48$, $p < 0.001$) and nitrite-N ($r = 0.56$, $p < 0.001$) (Supplementary Fig. 9), was also identified as a key driver of Bdellovibrionota, Nitrospirota, and Planctomycetota relative abundance, accounting for 4.89, 4.51, and 1.33% of variance, respectively (Fig. 6E). These results suggest that WWTP effluent, which introduces organic nutrients into the watercourse, may impact the dynamics of biofilm bacterial communities, with possible implications for microbial nutrient cycling and water quality[14].

The River Continuum Concept (RCC)[63] describes how continuous physical gradients along a river's course, from headwaters to mouth, shape its biological and chemical properties. Low-order streams are shallow, narrow and strongly influenced by shading and allochthonous input from vegetation. As stream order, and therefore river width and depth, increase towards the mouth, autochthonous inputs play a progressively larger role[63]. Successional shifts in surface water bacterial communities along the river continuum have been well-documented within individual catchments[64–66]. This trend extends to the national-scale, where stream order was identified as the most significant environmental driver shaping bacterial communities in the surface water of rivers across North America[25], ranging from stream order 1 to 12. In contrast, stream order within the RSN, which ranged from 1 to 8, was not a dominant driver of bacterial community

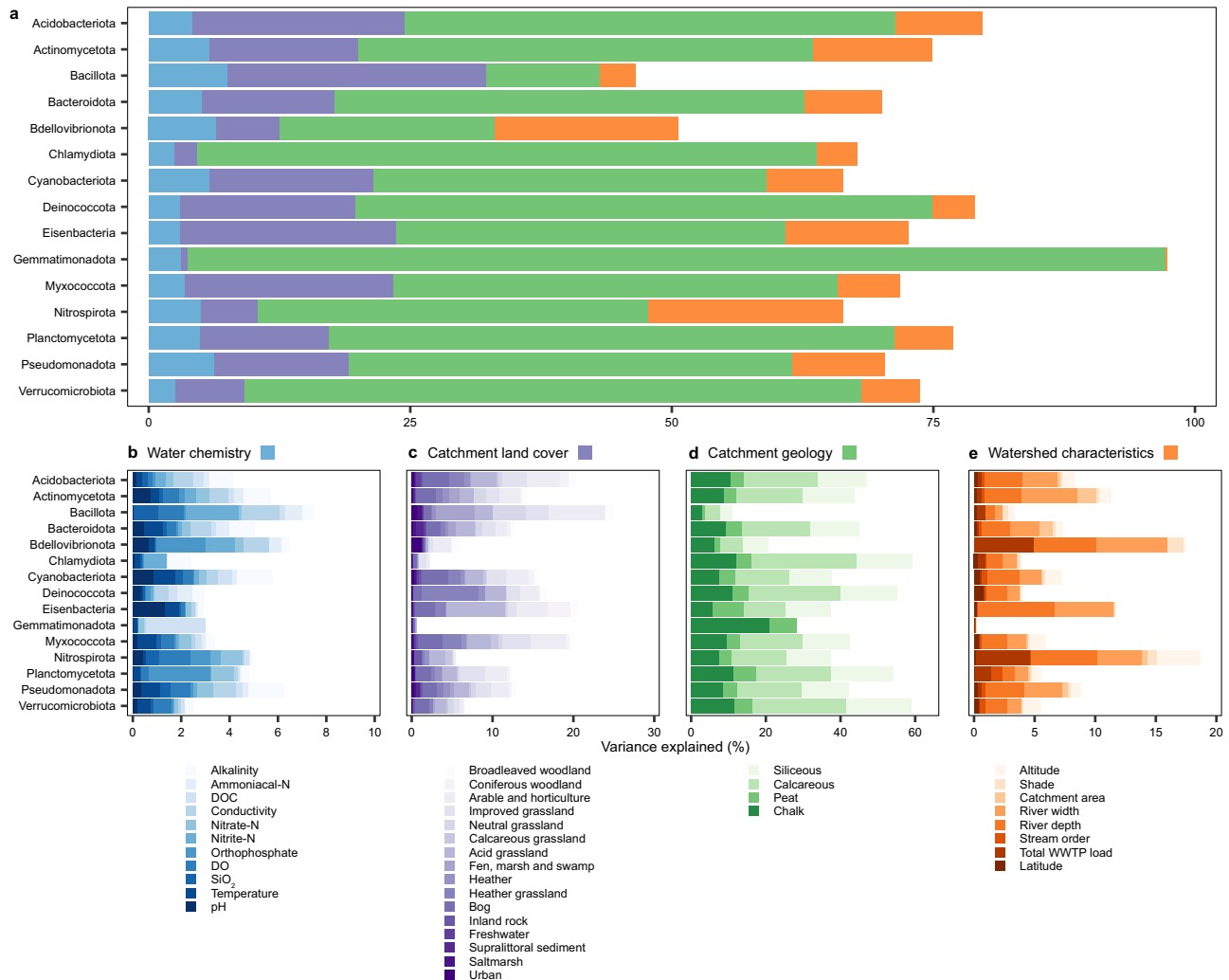

**Fig. 6 | Environmental drivers of river biofilm bacterial communities. a** Total variance explained by water chemistry, upstream catchment land cover, upstream catchment geology, and watershed characteristics, and variance explained by **b** water chemistry variables, **c** upstream catchment land cover types, **d** upstream catchment geology types, and **e** watershed characteristics. Variance explained for each bacterial phylum is calculated as a mean of variance explained across individual MAGs. Source data are provided as a Source Data file.

composition in river biofilms across England, explaining less than 1% of variance for each bacterial phylum (Fig. 6e). Associated physical factors such as river depth and width explained up to 6.31 and 5.81%, respectively, but shading accounted for less than 1.32% of bacterial community composition. Bacterial communities within stream biofilms from a pre-alpine catchment in Austria (stream order 1–5) were also found to show patterns contradictory to the RCC[67]. These contrasting findings between river surface water and biofilm bacterial communities suggest that successional changes along the river continuum may affect these distinct microbial communities differently. While variation in surface water bacterial communities can be primarily explained by the RCC, local factors such as geology, land cover, nutrients and WWTP load appear to be more influential in shaping river biofilm communities. However, the smaller range of stream orders in the English RSN, compared with those investigated in North America, may partially contribute to the weaker relationship observed between stream order and biofilm community composition. Further studies of biofilm bacterial communities in larger rivers and comparisons between co-located water column and biofilm samples are required to fully understand the influence of the RCC on freshwater bacteria. Seasonal fluctuations could also influence river biofilm community composition, functional potential, and the relative importance of

environmental drivers, though seasonal effects have yet to be comprehensively explored at a large-scale. Furthermore, cross-domain biotic interactions may also be an important driver of biofilm bacterial community dynamics and require further study[54,67].

## Conclusion

Rivers are valuable, biodiverse habitats and their microbial communities, particularly those within biofilms, are critical to ecosystem functioning, biogeochemical cycling, pollutant degradation, and maintaining water quality[2,6]. By leveraging comprehensive national-scale sampling coupled with high-resolution environmental data, this study provides novel insights into the biogeography, taxonomic diversity, metabolic potential, and functional roles of river biofilm microbial assemblages.

River biofilms are revealed to be dominated by high-occupancy generalist bacteria, most notably members of Pseudomonadota, Cyanobacteriota and Bacteroidota, reflecting the interconnectedness of river ecosystems that facilitate microbial dispersal across the river network and colonisation by generalist taxa. These results further highlight the significant metabolic and functional versatility of river biofilms, which harbour a diverse array of genes supporting carbon, nitrogen and sulfur cycling, the utilisation of a broad range of organic

compounds, the transportation of organic pollutants, and various trophic strategies. Their capacity to perform diverse biochemical roles reflects niche adaptation to the biofilm microenvironment and provides resilience in dynamic river ecosystems, allowing them to sustain key biogeochemical processes under fluctuating or unfavourable conditions.

The RCC is a strong driver of surface water bacterial communities[25,64–66]. However, this framework is less applicable to river biofilm communities. Instead, upstream catchment geology and land cover emerge as the primary determinants of biofilm communities, which likely shape community composition through their long-term effects on the physicochemical conditions of the water column and importantly the biofilm microenvironment. Furthermore, anthropogenic activities, including changes in land use and the release of WWTP effluents and pollutants into rivers, may significantly impact biofilm community dynamics. River biofilms are valuable bioindicators of ecosystem health[58], and our findings provide baseline data that may support water quality monitoring and management, and, importantly, explain resilience to environmental change observed in river biofilm bacterial communities.

## Methods

### Sample collection
A total of 450 river biofilm samples were collected over a three-year period (2021–2023) from 146 sites within the EA's RSN programme. The sampling sites were selected using a randomised, spatially balanced design to ensure representative and unbiased spatial coverage across England[68]. A total of 174 samples were collected in 2021, 178 in 2022, and 98 in 2023, with 62 sites being sampled in all three years. Biofilms were collected from the river benthos by scraping biofilm-covered stones from the riverbed at each site with a sterile toothbrush and deionised water[69]. At sites where stones were not accessible, biofilms were collected by macrophyte scrape. The suspended biofilm was transferred to a 15 ml tube and preserved in the field in 5 ml nucleic acid preservation buffer (3.5 M ammonium sulphate, 17 mM sodium citrate and 13 mM EDTA)[70]. Samples were immediately transported to the EA's National Laboratory at Starcross, Exeter where they were concentrated by centrifugation and frozen. Samples were then transported to the UK Centre for Ecology & Hydrology (UKCEH), Wallingford on dry ice and stored at −20 °C.

### Water chemistry
Water chemistry variables, including water temperature (°C), pH, alkalinity to pH 4.5 as $CaCO_3$ (mg $L^{-1}$), conductivity, and the concentration of chloride ion (mg $L^{-1}$), dissolved oxygen (DO, mg $L^{-1}$), dissolved oxygen saturated (mg $L^{-1}$), dissolved organic carbon (DOC, mg $L^{-1}$), total phosphorus (TP, mg $L^{-1}$), orthophosphate (mg $L^{-1}$), total nitrogen (TN, mg $L^{-1}$), total oxidised nitrogen (TON, mg $L^{-1}$), nitrate-nitrogen (nitrate-N, mg $L^{-1}$), nitrite-nitrogen (nitrite-N, mg $L^{-1}$), ammoniacal nitrogen (ammoniacal-N, mg $L^{-1}$), ammonia-N unionised (mg $L^{-1}$), and reactive $SiO_2$ (mg $L^{-1}$), were measured using surface water samples collected from each sampling site. All water chemistry data is available at: https://environment.data.gov.uk/water-quality/view/landing. For each variable, a mean was calculated from up to 5 independent measurements taken on separate occasions at each site across a 3-month period up to and including the day of biofilm sampling and is provided in Supplementary Data 1.

### Upstream catchments
The upstream catchment of each RSN site was determined using 10 m flow direction and accumulation grids and the sampling points which were snapped to the maximum flow accumulation cell within 50 m of its location. The digital elevation model has a 10 m resolution and was based on NEXTMap which was derived from airborne Interferometric Synthetic Aperture Radar (IFSAR) and is available on the UKCEH

Catchment Management Modelling Platform (https://catalogue.ceh.ac.uk/cmp/documents). The area of the derived shapefile was then used as the area of the upstream catchment (km²). The upstream catchment shapefiles were intersected with the 10 m scale 2020 and 2021 UKCEH land cover maps available at: https://www.ceh.ac.uk/data/ukceh-land-cover-maps, and the British Geological Survey (BGS) geology map available at: https://www.bgs.ac.uk/datasets/bgs-geology-250k. The percentage of the upstream catchment area covered by each land cover type or geology was calculated for each sampling site.

### Watershed characteristics
Strahler numbers for the branches of the rivers where sampling sites are located were determined by spatially joining the GRTS points with the 2021 OS Open Rivers Network layer (https://www.ordnancesurvey.co.uk/products/os-open-rivers). To provide an estimate of river depth and width at sampling sites, sampling points were intersected with the gridded 1 km² physical river characteristics for the UK v2 dataset (https://doi.org/10.5285/8df65124-68e9-4c68-8659-1c6b82c735e9).

To determine the number of wastewater treatment plants (WWTPs) in the upstream catchments of each site and the total population equivalent loads, a spatial intersect was performed between the upstream catchments and a dataset of WWTPs in England and Wales with their population equivalent loads mapped to discharge location (https://www.data.gov.uk/dataset/0f76a1c3-1368-476b-a4df-7ef32bfd9a8b/urban-waste-water-treatment-directive-treatment-plants). Count and sum functions were used to provide the number of WWTPs and their total population equivalent loads respectively.

Average shading from landforms and surface model objects was determined for each sampling by spatially joining the sampling points with a 25 m buffer zone to the Environment Agency Keeping Rivers Cool relative shading map available at: https://data.catchmentbasedapproach.org/maps/theriverstrust::riparian-shade-england.

### DNA extraction
Each biofilm sample was defrosted and briefly vortexed to homogenise the sample. The Quick-DNA fecal/soil microbe kit (Zymo Research, California, U.S.) was used to extract DNA from 100 μL of sample with the following amendments to the manufacturer's protocol to maximise DNA yield: Zymo DNA/RNA shield was used as the lysis buffer, samples were lysed at 20 Hz for 20 min using the TissueLyser II (Qiagen, Germany), and 20 μL of proteinase K was added to the lysate prior to incubation at 65 °C for 20 min[71]. Purified DNA was eluted in 100 μL of elution buffer. The purity of extracted DNA was checked using the NanoDrop 8000 spectrophotometer (Thermo Fisher Scientific, MA, U.S.). The concentration of DNA was measured using the QuantiFluor ONE dsDNA kit (Promega, Madison, WI, U.S.) with a BioTek Cytation 5 imaging reader (Agilent Technologies, California, U.S.) according to the manufacturer's protocol. The DNA was stored frozen at −70 °C for long-term archiving at UKCEH, Wallingford.

### Metagenomic sequencing
Extracted DNA was sent to Novogene UK for library preparation and 2 × 150 bp shotgun metagenomic sequencing on an Illumina NovaSeq 6000 with an S4 flow cell to achieve a sequencing depth of at least 10 Gb raw data per sample. Between 2.85 million and 188.26 million raw reads were generated per sample, with a median of 72.16 million raw reads per sample.

### Metagenomic data processing
Illumina adaptor sequences were trimmed, and metagenomic reads were filtered to a minimum quality score of 25 using Trim Galore v0.6.5 (https://github.com/FelixKrueger/TrimGalore). Reads mapping to the human reference genome (GRCh38; downloaded on 2023-11-16 from

https://ftp.ncbi.nlm.nih.gov/refseq/H_sapiens/annotation/GRCh38_latest/refseq_identifiers/GRCh38_latest_genomic.fna.gz) were removed. After quality filtering and trimming, between 4.28 million and 90.17 million reads were obtained per sample, with a median of 36.37 million reads per sample. To determine the overall profile and percentage of archaea, bacteria, and eukaryotes in the dataset, singleM v0.16.0[72] was run on the pre-processed reads. MultiQC v1.17[73] was used to process all filtered reads to obtain basic statistics per sample. Megahit v1.2.9[74] was used to assemble the reads into contigs, which were subsequently used for binning, functional annotation, and downstream analyses. For functional analysis, open-reading frames (ORFs) were predicted using Prodigal v2.6.3[75], which were fed into EggNOG-mapper v2.1.9[76] for functional annotations via the built-in v6.0 database. The ORFs were used in featureCounts[77] to generate per-gene coverage, while Kraken2 v2.1.2[78] and Bracken v2.6.0[79] were used to taxonomically annotate the contigs, using the PlusPFP (https://benlangmead.github.io/aws-indexes/k2) database, which includes RefSeq protozoa, fungi, plant, archaea, bacteria, virus, plasmid, and human sequences. For stringency, a 0.7 confidence threshold was used in Kraken2. The contigs were also used to obtain metagenome-assembled genomes (MAGs), by dereplicating bins obtained via MetaBAT v2.15[80], MetaBinner v1.4.3[81] and CONCOCT v1.1.0[82] using dRep v3.2.2[83]. The completion and contamination of the dereplicated bins were estimated with CheckM2 v1.2.2[84], alongside taxonomic annotation using the comprehensive and phylogenetically consistent prokaryotic genome taxonomy database, GTDB-Tk v2.3.2[85]. Species were delineated using standard GTDB-Tk thresholds of ≥95% average nucleotide identity (ANI) across ≥65% of the genome alignment fraction.

MicroTrait v1.0.0[86] was used to extract fitness traits from the MAGs, based on protein family sequence similarities and Kyoto Encyclopedia of Genes and Genomes (KEGG) orthologs (KOs). The metabolic and biogeochemical traits and pathways of the MAGs were also analysed using METABOLIC v4.0, which integrates numerous annotation databases, including KEGG and curated hidden Markov model (HMM) databases[87]. The metabolic pathways were further assessed via metabolisHMM v2.22, which uses curated HMMs for functional annotations of metabolic characteristics spanning carbon, nitrogen, sulfur, and hydrogen cycling pathways[88]. MicrobeAnnotator v2.0.5 was used to annotate the MAGs using the -light option against the Kofam and Swissprot databases[89]. Biofilm associated KOs were then identified according to the BBSdb of bacterial biofilm-associated proteins[90] supplemented with additional KOs found in KEGG pathways such as quorum sensing (map02024), flagellar assembly (map02040), chemotaxis (map02030), and biofilm formation (map02025, map02026, map05111). The workflow was implemented using the Snakemake workflow management system v7.8.2[91], and is available at: https://github.com/amycthorpe/metag_analysis_EA for preprocessing the metagenomic reads to MAG assembly. The analysis workflow, including dereplication, taxonomy assignment and the implementation of microTrait, METABOLIC, and metabolisHMM, is available at: https://github.com/amycthorpe/EA_metag_post_analysis.

## Data analysis

Using contamination and completeness statistics estimated with CheckM2, the MAGs were categorised as near-complete (>90% complete and <5% contamination) or medium-quality (>70% complete and <10% contamination)[92], with remaining MAGs that did not meet the thresholds categorised as low-quality. Medium and high-quality MAGs were retained for downstream analysis. A phylogenetic tree was constructed using the maximum likelihood method with the ggtree R package v3.12.0[93] using the multiple sequence alignment from the genome taxonomy database toolkit analysis (GTDB-Tk). For biogeographic mapping, relative abundance was determined for each MAG, summed at the phylum level and plotted according to site latitude and longitude using rnaturalearth R package v1.10.9000[94].

A Bray–Curtis beta diversity dissimilarity matrix was generated based on MAG relative abundance using the vegan R package v2.6.10[95]. The relationship between the dissimilarity matrix and geographic distance between samples was investigated with a linear regression. Non-metric multidimensional scaling was performed based on the dissimilarity matrix. Environmental variables were fitted to the ordination space using the vegan envfit() function with 999 permutations. To investigate sample clustering within the ordination space by dominant geological drivers, the log ratio of the proportion of the upstream catchment represented by calcareous to siliceous geologies was calculated.

For occupancy and relative abundance analyses, mean relative abundance was calculated for each MAG across all samples, and the number of samples each MAG was present in was used as a measure of occupancy. Levins' niche breadth ($B_N$) was calculated for each MAG using the MicroNiche R package v1.0.0[96]. This metric quantifies the proportional similarity of resource use, where a $B_N$ closer to one is indicative of a more uniform distribution and therefore a broad niche and generalist lifestyle, while a $B_N$ closer to zero indicates a narrower niche with a preference for specific conditions[96]. MAGs with a $B_N$ greater than the median $B_N$ (0.03) were categorised as generalists, and those with a $B_N$ less than the median were categorised as specialists. All MAGs were above the recommended limit of quantification threshold of 1.65. The percentage of generalists and specialist MAGs was calculated for each bacterial phylum. The presence of metabolic and functional traits identified for each MAG using METABOLIC, metabolisHMM, and microTrait were weighted by mean relative abundance and summed at the phylum level and genus level. Biofilm-associated KOs identified with MicrobeAnnotator were normalised by coding density per MAG.

Variance partitioning was used to identify the environmental drivers of bacterial community composition at the MAG level. A linear mixed model using maximum likelihood estimation was fitted to the matrix of MAG relative abundance to estimate the contribution of each environmental variable to community variation using the variancePartition R package v1.34.0[97]. Due to low variance for some MAGs, variance partitioning was performed based on a subset of 340 MAGs. Samples ($n = 450$) were filtered to complete observations according to environmental data availability ($n = 401$). Pearson correlations (two-sided) were computed using the cor() function in R to investigate correlation among the environmental variables and with bacterial phyla on the subset of 401 samples with complete observations. To optimise variance partitioning model performance, some measured water chemistry parameters, including total phosphorus, were not included in the analysis due to significant co-correlation ($r > 0.80$, $p < 0.001$) with other, more biologically available measures (e.g., orthophosphate, Supplementary Data 5). Variance explained per phylum was calculated as a mean of variance explained per MAG. Data analysis was performed in R v4.4.0[98], and R scripts for data analysis and visualisation are available at: https://github.com/amycthorpe/biofilm_MAG_analysis.

## Reporting summary

Further information on research design is available in the Nature Portfolio Reporting Summary linked to this article.

# Data availability

The metagenomic data generated in this study have been deposited in the European Nucleotide Archive (ENA) at EMBL-EBI under accession number PRJEB85861. Sample accession codes and all the data generated, including the environmental metadata associated with each sample, MAG coverage, taxonomy, and CheckM2 statistics, outputs from METABOLIC, metabolisHMM, and microTrait, and the niche breadth index, variance partitioning, and correlation analysis results are available on Zenodo at: https://doi.org/10.5281/zenodo.14762144. Source data are provided with this paper.

## Code availability

Snakemake workflows used to process and analyse the metagenomic data are available on GitHub (https://github.com/amycthorpe/metag_analysis_EA, https://github.com/amycthorpe/EA_metag_post_analysis) and archived on Zenodo[99,100]. R scripts for data analysis and visualisation are available on GitHub (https://github.com/amycthorpe/biofilm_MAG_analysis) and archived on Zenodo[101].

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

## Acknowledgements

Thank you to Dr Jonathan Porter, Sean Butler, and Alan Wan from the Environment Agencies National Monitoring Laboratories for their support collating and shipping the biofilm samples for further processing. This work was funded by the Environment Agency under research project SC220034. A.C.T., S.B.B and D.S.R. were supported by Natural Environment Research Council (NERC) grant NE/X015947/1. K.W. and J.W. were supported by NERC grant NE/X015777/1. The authors acknowledge the support of the Biotechnology and Biological Sciences Research Council (BBSRC), part of UK Research and Innovation; Earlham Institute Strategic Programme Grant Decoding Biodiversity BB/X011089/1 and its constituent work package - BBS/E/ER/230002C (Decode WP3 Linking Fine-Scale Microbial Diversity to Ecosystem Functions).

## Author contributions

A.C.T.: conceptualisation, data curation, investigation, formal analysis, visualisation, writing – original draft, review and editing. S.B.B.: conceptualisation, data curation, formal analysis, writing – review and editing. J.W.: conceptualisation, data curation, writing – review and editing. L.H.H.: data curation, writing – review and editing. K.W.: conceptualisation, supervision, writing – review and editing. D.S.R.: conceptualisation, supervision, writing – review and editing.

## Competing interests

The authors declare no competing interests.
