## [Transparent Peer Review file · Nature Communications]

National-scale biogeography and function of river and stream bacterial biofilm communities

Corresponding Author: Dr Amy Thorpe

Version 0:

Reviewer comments:

Reviewer #1

(Remarks to the Author)

The authors present an ambitious study that investigates microbial taxonomic and function diversity of river biofilms at a broad geographic scale, highlighting their ecological importance and response to environmental change. This is timely given the increasing pressures of climate and land use change. Genome size, GC content, bacterial community composition, and predicted functional traits were assessed as a function of different environmental drivers, spatial distribution, watershed characteristics, and niche breadth. The authors found biofilms were dominated by generalists, including many novel taxa, with geology, land cover, and nutrients explaining ~90% of community variation. Given the complexity of river biofilms, and the factors that regulate them, the general conclusions from this work support the importance of river biofilms and their ecological roles.

The scope of this study is impressive, and the authors clearly demonstrate their expertise in managing and interpreting complex environmental and genomic datasets. The use of variancePartition for modeling environmental drivers of microbial composition, niche breadth estimation, and the integration of trait-based approaches across geographic space adds further depth to the study. This integrative approach provides valuable insight into the ecological roles and resilience of microbial biofilms in river systems.

However, despite the study's broad geographic scope, the results are largely descriptive and fall short of advancing understanding of the genes, metabolic pathways, or specific environmental pressures that regulate river biofilm composition and function. While novel in scale, the results offer few new mechanistic insights or conceptual frameworks, providing limited advancement to the field of microbial ecology.

Major comments:

The dataset used in this study is extensive and presents a valuable opportunity to advance our understanding of biofilm community structure and function across river systems. However, the manuscript would benefit from a more cohesive narrative and a clearer articulation of the hypotheses and research questions. Currently, the results are largely descriptive, and although some broad patterns are reported (e.g., gene presence, taxonomic composition at the phylum level), the manuscript does not fully capitalize on the richness of the dataset. Moreover, it lacks structural cohesion and a clear narrative to appeal to a broad readership.

Several areas could be strengthened. First, only a subset of available environmental variables (e.g., orthophosphate only rather than including total phosphorus) are analyzed, limiting the interpretability of nutrient-driven patterns. Moreover, since geology accounts for the largest proportion of variance, including metrics like CaCO₃ (i.e., alkalinity) would also be relevant—does alkalinity in rivers depend on surrounding geology? While the authors highlight geology as a strong driver (e.g., calcareous vs. siliceous substrates), it's unclear whether rivers were clustered or compared along those axes to support this conclusion more directly. Similarly, while geology and land cover are identified as primary drivers of biofilm communities, their influence likely occurs through changes in water chemistry. However, water chemistry explained little variation in community composition, raising the question of whether this relationship is adequately captured, or if other factors are more directly influencing biofilm composition. This could use more clarification and discussion.

Second, analyses are restricted to phylum-level, which may obscure ecologically meaningful variation at finer taxonomic levels. Incorporating genus-level trends could yield new insights, especially when paired with genome-resolved functional

traits. Although the authors emphasize functional potential (e.g., genes for carbon and nitrogen utilization), variance partitioning shows only modest explanatory power for environmental variables. More discussion on microbe-microbe interactions within the biofilm matrix or other potential variables responsible for the presence of these genes is warranted.

In short, the manuscript gestures at the importance of river biofilms and their functional capacity, but the analysis and interpretation stop short of delivering a clear conceptual advance. Many of the discussion points are overly broad and lack direct relevance to the results, which weakens their impact and clarity in relation to the study's findings. With stronger framing, more targeted use of the data, and a sharper interpretation of key findings, this study could make a more compelling contribution.

Minor comments:

L1: The title should reflect the main findings of the paper. Currently, it reads too broad. Please refine.

L26: The sum total variance here is misleading since nutrients accounted for a smaller contribution compared to geology and land cover.

L31: This is mentioned in the provided citation, the relevance/importance of fungi in rivers systems should be noted, see <https://pmc.ncbi.nlm.nih.gov/articles/PMC4601154/pdf/emss-63879.pdf>

L35: Aquatic plants? Litter from riparian vegetation? Please clarify.

L36: Microbe-microbe interactions (quorum sensing, cooperation/competition) also occur within the biofilm matrix and contribute to composition and function.

L41: How could these results help inform management? No mention of application other than it could, rather than how.

L43: Microbe-microbe interactions? Macroinvertebrate grazers?

L47: Good to include a mention of biotic interactions.

L49: How is this limited? In what ways?

L51: What obstacles make this challenging? Moreover, <https://www.nature.com/articles/s41522-021-00251-2> discusses the advantage of forming biofilms in mitigating against environmental stressors as well as antagonistic microbe-microbe interactions—could biofilms be more resistant to change compared to water column, sediment or other sink locations in rivers?

L71: Did you control for interannual variability? Seasonality likely plays a big role see <https://journals.asm.org/doi/10.1128/AEM.72.1.713-722.2006>
<https://www.nature.com/articles/s41598-020-73293-9>
<https://www.nature.com/articles/s41586-024-08240-z>
<https://journals.asm.org/doi/10.1128/aem.72.1.713-722.2006>

L72: Why were these sites chosen relative to others? Were the same sites sampled over the 3 year period?

L78: Do you have total Phosphorus? This is a limiting nutrient important for microbial metabolism.

L85: This header should be tailored to the content of this section—primarily methods and results for GC association with env. variables. should be divided accordingly.

L88: Why? How is it different?

L95: Can this be specified at a higher taxonomic resolution? Fungi, algae, etc. Algae are a large carbon source for heterotrophic microbes in biofilms see <https://www.sciencedirect.com/science/article/pii/S0923250815000819?via=ihub>
The role of algae in biofilm function might explain some of the variation observed in this study.

L100: It is surprising that on average only 2 distinct taxa per sample were found (~1000 taxa across ~400 samples). As sequencing depth was impressive at nearly 70M reads per sample, inadequate depth does not seem to explain this. The authors should discuss this in the context of their argument that these biofilms harbor mostly generalists. However, this also highlights whether dereplication was appropriate for this dataset. This step prevents the authors from detecting population level variance that may have corresponded more strongly with the measurement of environmental parameters.

L108: Many microbial studies refrain from assigning bacterial taxa down to the species level. It is therefore unclear whether this finding that 95.1% of recovered MAGs represented previously uncharacterized species is truly a discovery of novel, undescribed taxa. This seems like it could have been caused by the choice of database or analysis pipeline.

L111: Please highlight why Phylum-level was chosen as opposed to other taxonomic levels. Family- or order-level might yield new insights, especially in conjunction with predictive function see Shah et al. 2024 <https://enviromicro->

L117: What type of rivers? Similar rivers to those in this study, globally? Would also be good to mention if these rivers are rain or spring-fed since you also mention glacial-fed rivers.

L121: This statement relates to my concern about dereplication. Demonstrating that GC content or genome size varies across broad phylogenetic groups is not new. However, it is conceivable that further analysis of this dataset at finer taxonomic levels might reveal exciting patterns between environmental parameters and GC content and/or genome size. This paragraph as a whole seems problematic because of the coarse taxonomic scale used for this analysis.

L129: Is there a citation for this? If the citation is in the following sentence then please merge since it reads as a stand-alone.

L130: Can you clarify the statement of stability? This reads as if nutrient-rich environments are less stable?

L132: What hypothesis? That genome-size may confer an advantage in nutrient-specific environments? Please state more clearly.

L136: Have you investigated other genomic parameters beyond gene size and %GC? see Shah et al. 2024 for work on ABC transporters and other adaptations across a large number of freshwater lakes <https://enviromicro-journals.onlinelibrary.wiley.com/doi/10.1111/1462-2920.16634>
Grazers have been found to influence biofilms, the idea of aquatic macroinvertebrates as a source or dispersal mechanism of biofilms could be highlighted. Moreover, temperature, light, as well as surrounding riparian vegetation could also explain variation based on certain genes/metabolic pathways. If you have the data, this would be good to include. see Borton et al. <https://doi.org/10.1038/s41586-024-08240-z>

L159: Would be good to bring in the Baas Becking hypothesis here for context.

L181: This finding suggests that dispersal is high within river systems, but are you seeing distinct compositions across rivers? Within rivers between upstream and downstream?

L189: What is known so far? The authors mention that "X is limited, or unknown" but do not provide much commentary or mention on the knowledge gaps themselves.

L192: Semantics, but "contributed" implies activity but this study only addresses functional capacity. Descriptions should reflect this (e.g., "contribute to X based on genes encoding for X").

L194: A few more words are needed here to explain methodology. Where did these cycles come from? KEGG pathway database? Further, many of the findings in this paragraph are descriptive and do not differ from expectations based on previously described freshwater bacteria.

L212: With no association with the environmental drivers of your study, how does this relate? This seems to be a list of all annotated genes without a framework of the broader relevance of these genes. See <https://www.nature.com/articles/s41522-021-00251-2> which also mentions that the biomatrix itself can serve as a resource for bacterial communities. It would be good to include more discussion on how dissolved biopolymers from the matrix could perhaps buffer potential effects from nutrient fluctuations or disturbance.

L213: predicted functional capabilities?

L225: Penesyan et al 2020 <https://www.nature.com/articles/s41522-021-00251-2> explains that the stratification of river biofilms is dependent on location in the water column "shallow aquatic environments, microbial species are organised according to their metabolic and energetic properties" since you have depth data this could be useful to include. Moreover, could you clarify if this is related more to the env. variables of interest, or from the biofilm matrix?

L240: Cooperative and antagonistic interactions? Selective pressure of microbe-microbe interactions are also at play and should be mentioned. Did you notice community turnover across sampling years? How consistent are the communities over time?

L249: How precise are these genes for anthropogenic pollutants? It would be useful to clarify that these polycyclic aromatic compounds differ from naturally found aromatics, such as those comprising plant secondary metabolites.

L245: Do the rivers in this study have a history of pollution? If so, from what type (mining, agriculture, sewage, etc.)?

L251: Do you have a table of these genes?

L252: Similarly, can we be sure 'transport of metals' is related to industrial discharge rather than the transport of metals in natural systems without pollutants? (i.e. siderophores for Iron)

L266: Planktonic? Please clarify.

L274: Please provide more detail for the other characteristics.

L275: Please state which factors.

L281: Please identify which land cover types contributed most to composition.

L291: Have you explored if these metrics differ between rivers grouped by geology type? The dataset also contains CaCO₃, so did you find if concentrations vary among your sites?

L293: Perhaps a more precise phrase than 'water chemistry' is needed. The prior sentence discusses that pH and alkalinity do have important effects on biofilm communities. Water chemistry traditionally includes pH and alkalinity.

L296: The Methods state that biofilms were scrapped from stones and macrophytes. Please rework this statement given the locations from which biofilms were sampled (not only stones)—it does not seem probable that microbes would directly rely on stone-derived nutrients. It would be interesting, however, to see if P water chemistry varies by geology since soil and bank erosion also contribute to P in river systems. See this paper on geologic N inputs <https://www.nature.com/articles/27410>

L296: A bit of a stretch to include this unless you have chemistry data on the biofilm itself, otherwise should be stated with caution.

L320: Were there relevant methodological differences between the two studies? For example, it is unclear through this point in the manuscript whether the current study sampled biofilms of both organic and non-organic substrates.

L314: Do you find that cyanobacteria are more abundant in higher order streams? Or other photosynthetic Eukaryotes?

Figure 1: Presents a detailed overview of MAG quality, taxonomic novelty, and phylogenetic diversity, which is useful for assessing the scope of the dataset. However, it functions more as a summary of dataset characteristics rather than a figure that advances the narrative or supports a central hypothesis. Consider moving this figure to the supplement and using main-text figures to focus more directly on ecological or functional insights derived from the MAGs.

Figure 3: Legend needs to be more detailed.

Figure 4: Where did these pathways in a - c come from? These pathways seem to be overly simplified.

Figure 5: This is a nice summary of biofilm function, but the major conclusions to be drawn from this are unclear.

L577: Was spatial sampling across and within years randomized? Certain biases could have arisen in the dataset if certain regions/landscapes were sampled in a single year whereas a different landscape was sampled in a different year. If randomization was not done, then showing limited year-to-year variation in important parameters would be useful (i.e. were any of these years significantly lower in flow rate/drought?).

L581: Were any statistical comparisons made between biofilms collected from river stone versus those collected from macrophytes?

L582: This looks like a commonly used mixture to preserve RNA, not DNA. This should be corrected to avoid confusion and a citation should be included to justify this mixture works well as an alternative to premade solutions (i.e. RNALater).

L595: Measurements were taken during a three month period, which was prior to biofilm collection, but were these measurements taken exactly three months prior to sampling? If not, what range of timespans elapsed between nutrient measures and biofilm sampling? How well do these nutrient measurements correspond with the conditions that occurred at these sites during biofilm collection?

L649 and 658: Given this range of sample depths, median depth would be more informative than mean.

L689: These are well accepted standards, but should still be cited: <https://www.nature.com/articles/nbt.3893>

L695: Please provide a more detailed (brief) explanation of this. Helpful for a broader audience
See Finn et al 2020 <https://academic.oup.com/femsec/article/96/8/fiaa131/5863182>

L704: Please identify the LMM adjustment used.

Reviewer #2

(Remarks to the Author)

Reviewer #3

(Remarks to the Author)

The study "National-scale biogeography and function of river and stream bacterial biofilm communities" by Thorpe et al. provide a more comprehensive understanding of the ecological dynamics of freshwater microbes and their drivers at a global scale. More specifically they highlight (i) the microbial community composition of the river biofilm across the 450 samples in 126 National River Surveillance Network sites, (ii) metabolic potential of recovered 1014 MAGs in C, N and S cycling were illustrated, and (iii) role of environmental drivers on variation in community composition. The concept of the study was quite good and findings are interesting.

However, out of the total MAGs obtained, authors included the low-quality MAGs with more than 10% contamination in metabolic analysis (line 195). Recent studies are generally considering the MAGs with less than 10% contamination for analysis (Borton et al. 2024. A functional microbiome catalogue crowdsourced from North American rivers. *Nature*; Garner et al. 2023. A genome catalogue of lake bacterial diversity and its drivers at continental scale. *Nature Microbiology*; Bowers et al. 2017. Minimum information about a single amplified genome (MISAG) and a metagenome-assembled genome (MIMAG) of bacteria and archaea. *Nature Biotechnology*; Singleton et al. 2024. Microflora Danica: the atlas of Danish environmental microbiomes. *bioRxiv*). I suggest to re-do the analysis without low-quality MAGs. It can misinterpret the results in terms of metabolic potential due to contamination. It is quite confusing as a reader in some places authors used only high-quality MAGs (line 109), in some places all MAGs were included. Along with this author did not discuss about the genes related to biofilm formations in MAGs as well the overall discuss was not well written. There are few more questions need to be addressed to improve the quality of the results and their interpretations.

1- Authors need to remove the citation from the abstract.

2- Line 20: Are these 1014 MAGs high quality ? and classified up to species or strain level

3- Line 25: Environmental drivers, most notably geology, land cover, and nutrients, explained up to 90% of the variation in community composition. The value is quite high than previously reported average variation. Authors need to clarify this? Is it for overall community composition or for individual taxa?

4- There is no result and discussion heading in the Manuscript. Please add to separate the section from the introduction.

5- Line 78: Unit of measurement is not provided in "Extended Data Table 2". Summary of water chemistry data.

6- Line 671: Which tools authors had used for dereplication of bins?

7- In the section "Metabolic and functional potential of river biofilm communities", genes related to biofilm are not described. Freshwater system can be a source of antimicrobial resistance genes, it would be great if authors can find the MAGs carrying these genes and how their abundance vary across the river network. Authors could have described more about the bioremediation potential of these microbes.

8- Freshwater is considered as one of the source of GHG emission, therefore author can find out the genomic potential of these MAGs in GHG attenuation and how these biofilms can play an important role in GHG mitigation, if possible.

9- Authors can also perform the microbial source tracking of MAGs across the riverine network to understand the microbial movement from source to the sink and which are the seeding microbial members of the biofilm, if possible

10- Authors can perform differential abundant analysis for MAGs across the different sampling sites.

11- Authors need to improve the discussion in the whole manuscript as it is not up to the standard of the journal.

Version 1:

Reviewer comments:

Reviewer #1

(Remarks to the Author)

Thorpe et al have made a nice effort to address many of our minor concerns from the prior round of review. Unfortunately, we feel that several of our major concerns have not been adequately addressed by this revision. Nevertheless, we appreciate that the geographic scope of this dataset will without question provide excellent baseline data for many future studies. However, we think that many of the findings are overstated and unsupported, often missing key comparisons to controls and benchmarks, and so this manuscript would still require significant revision prior to consideration for publication.

A thorough description is provided of the biofilm community and genes within these taxa. However, with no quantitative comparisons of these results to the microbes inhabiting the water column, or other biofilm communities inhabiting other environments, there is no context for evaluating what is unique about these biofilm communities in terms of taxonomy, genomic characteristics or functional potential. Thus, the majority of results reported in this manuscript are descriptive and fall short of advancing understanding of the field of microbial ecology. For example, a sizable portion of the manuscript is dedicated to comparing GC content and genome size of different bacterial phyla. However, it is well appreciated in the field that such traits vary across the tree of life and the authors have not directly responded to this concern. How different are these features in their samples vs. those from a broad range of samples? Further, the authors argue that given the presence of many genes within the C, N and S cycling pathways and that these communities are vital to the cycling of these nutrients in rivers. We may be misinterpreting their statements, but in essence our understanding is that they are arguing that the presence of C, N, and S cycling genes is a surprising finding, when of course many microbes sequenced from many sources would have genes involved in C/N/S cycling given the fundamental importance of these genes to life. How does the occurrence of genes within these pathways differ from any other complex environmental community of bacteria? Are genes in these cycles over-represented in riverine biofilms compared to the null expectation (such as in contrast to the occurrence of these genes in surface water communities)? These controls are essential in order to make any substantive conclusion

about the role of riverine biofilms in C/N/S cycling, and the authors should be able to make these comparisons using publicly available data sets. The same holds for claiming the value of riverine biofilms in metal transport, as iron transport, for example, is essential for most life.

The manuscript makes large claims about the critical role of riverine biofilms in ecosystem function and “modulating fluxes of greenhouse gases in river ecosystems” (L261). It is not possible to conclude from metagenomic data alone that these communities have a sizable effect on greenhouse gas fluxes. Doing so requires in-situ measurements or mesocosm studies in which water quality changes are measured between treatments with and without biofilm communities, or contrasting measures when biofilm communities vary in taxonomic composition. At a minimum, measures of gene expression are needed.

Overall, we think this manuscript is still of high potential interest, because the results presented in Figure 6 are very interesting. Our suggestion was to focus on these results and perhaps pare down the other data presented unless these key control analyses are added. A stronger manuscript could be developed that focuses specifically on these results of how various environmental parameters predict biofilm composition and function. This is an outstanding dataset, but a sizable amount more needs to be done to elevate the interpretability and impact of these results.

Additional Specific Concerns

Title: there is no reference to a quite interesting aspect of this paper--- environmental parameters are predictive of biofilm biogeography

L121. Not following how a dataset with 311 genera has only 47 species?

L133. Based on the information currently provided, we are still not fully convinced of their argument that 94.4% of recovered MAGs represented previously uncharacterized species. Could you provide mean ANIs between your MAGs and the closest hit to the GTDBK database? As bacterial ‘species’ are an area of continued debate, more clarity of exactly how species are being defined is needed.

L152. Lack of context. If one were to look at Myxococcota MAGs in a range of environments, would they all have a large genome with high GC content? Unclear if this is an advance in understanding about the Myxococcota in general, or those specific Myxococcota that happen to inhabit riverine biofilms.

L160. Would be useful to add reviews about genome streamlining theory (Giovannoni et al. 2014).

L167-176. The manuscript is strengthened by this addition of genes involved in biofilm formation but again lacks sufficient context. Are these same genes universally found in other biofilms? In other biofilms, has it been found that “these traits were distributed across MAGs spanning diverse taxonomic groups?” Or is this a unique finding to riverine biofilms?

L210. The extra information regarding niche breadth is appreciated, but I am concerned this may be circular logic. If certain taxa are found in nearly all samples (i.e. high occupancy) in a sample set collected across a wide range of habitat types, would these taxa not by definition need to tolerate a wide range of habitat types found in the sample set and thereby be considered generalists? More discussion on how to interpret these results would be helpful.

L214. While a weak relationship between community dissimilarity and geographic distance could indicate high dispersal, is this the only possibility? If dispersal rates were low but evolutionary change was also low, or pressures on community composition/species sorting were uniform across space, would we not also see a weak relationship?

L277-280. These inferences are not supported by results in this study. There is no microspatial component to the sampling of these biofilms, at least as reported. So, it seems that the authors assume their diversity measures are a consequence of stratification. Authors should reduce causal language and remove statements that are not drawn from the presented results. Making such inferences could be acceptable if it is made clear that future studies are needed to prove causality.

L296-298. It is challenging to see how these claims are supported with no reference samples to compare to. Are these genes associated with stress tolerance strategies found in all bacteria? Are these genes significantly more prevalent in riverine biofilms, than for example the surface water communities?

L310-311. Certain metals, such as Iron, are essential for cellular growth. As a consequence, most bacteria have means of transporting such metals, such as siderophores. To make a claim that this high prevalence of metal transporters is due to transporting and transforming harmful pollutants, the authors must provide adequate context. Is this “99.39% of MAGs” significantly greater than the proportion of MAGs found in other systems to have these same genes involved in transport of metals?

L382. This statement needs to be moderated. This study does not measure how biofilms with certain community compositions versus others change nutrient cycling or impacts water quality.

L399-400. Comparison to the North American dataset is certainly warranted, but caution is needed when drawing interpretations between these two studies because there are multiple variables that differ. For example, the NA study

includes much larger rivers, which could have driven the results in this study that stream order was predictive. So, we do not think it is a valid argument that stream order shapes surface water bacterial communities but does not shape riverine biofilm communities. There are confounding variables between the two studies that would need to be addressed before such a claim could be made.

L418-421. It is difficult for a reader to appreciate how this is a 'diverse array of genes' and functions without any reference point. How does this compare to complex environmental bacterial communities in surface waters? Or other biofilms sampled in other environments? Again, here is where a contrast to bacterial communities in the surface waters could go a long way towards demonstrating why these riverine biofilms are potentially unique in their contributions to riverine function.

L433-435. Seems like an overreach. No measurements were taken for how biofilms alter water quality or public health. The link to biodiversity conservation is unclear. Making such claims necessitates functional measurements of how biofilms alter water quality.

Reviewer #2

(Remarks to the Author)

Reviewer #3

(Remarks to the Author)

The authors have comprehensively responded to all the comments, resulting in substantial improvements to the manuscript.

Version 2:

Reviewer comments:

Reviewer #1

(Remarks to the Author)

We think that Thorpe et al. have greatly improved their manuscript by providing additional context for many of their key results. We appreciate the careful revisions to incorporate new literature and draw comparison of their findings to those from biofilms in other environments, as well as to other riverine biofilm and water column samples from different locations. The authors have expanded upon key discussion between their work and the North American river dataset and have adequately revised their conclusions to avoid over-interpretations. We have no remaining concerns.

Reviewer #2

(Remarks to the Author)

Response to Reviewers

Manuscript ID: NCOMMS-25-17856A

Title: National-scale biogeography and function of river and stream bacterial biofilm communities

Our responses to each individual comment are given below in blue text. Tracked changes have been made in the revised manuscript, and quoted line numbers correspond to the clean 'no mark-up' version of the revised manuscript.

Reviewer #2 (Remarks to the Author):

Reviewer #1 (Remarks to the Author):

The authors present an ambitious study that investigates microbial taxonomic and function diversity of river biofilms at a broad geographic scale, highlighting their ecological importance and response to environmental change. This is timely given the increasing pressures of climate and land use change. Genome size, GC content, bacterial community composition, and predicted functional traits were assessed as a function of different environmental drivers, spatial distribution, watershed characteristics, and niche breadth. The authors found biofilms were dominated by generalists, including many novel taxa, with geology, land cover, and nutrients explaining ~90% of community variation. Given the complexity of river biofilms, and the factors that regulate them, the general conclusions from this work support the importance of river biofilms and their ecological roles.

The scope of this study is impressive, and the authors clearly demonstrate their expertise in managing and interpreting complex environmental and genomic datasets. The use of variancePartition for modeling environmental drivers of microbial composition, niche breadth estimation, and the integration of trait-based approaches across geographic space adds further depth to the study. This integrative approach provides valuable insight into the ecological roles and resilience of microbial biofilms in river systems.

However, despite the study's broad geographic scope, the results are largely descriptive and fall short of advancing understanding of the genes, metabolic pathways, or specific environmental pressures that regulate river biofilm composition and function. While novel in scale, the results offer few new mechanistic insights or conceptual frameworks, providing limited advancement to the field of microbial ecology.

We would like to thank the reviewers for taking the time to read and review our manuscript, and for their positive comments and insightful suggestions, which we have incorporated into our revised manuscript or clarified as necessary.

Major comments:

1.1. The dataset used in this study is extensive and presents a valuable opportunity to advance our understanding of biofilm community structure and function across river systems. However, the manuscript would benefit from a more cohesive narrative and a clearer articulation of the hypotheses and research questions. Currently, the results are largely descriptive, and although some broad patterns are reported (e.g., gene presence, taxonomic composition at the phylum level), the manuscript does not fully capitalize on the richness of the dataset. Moreover, it lacks structural cohesion and a clear narrative to appeal to a broad readership.

We thank the reviewer for this comment and as suggested have included research questions to better capture the narrative of our research in lines 93-106. Clearly stating the research questions that align with the structure of the results and discussion provides a stronger framework and emphasises the key novel insights gained from the dataset and their wider relevance. The revisions made throughout the manuscript in response to the suggestions below have expanded and improved the depth of our results, discussion, and conclusions.

1.2. Several areas could be strengthened. First, only a subset of available environmental variables (e.g., orthophosphate only rather than including total phosphorus) are analyzed, limiting the interpretability of nutrient-driven patterns.

Metadata variables were selected based on data completeness, co-correlation and ecological relevance to ensure robust analysis. The variance partitioning model fails if many strongly co-correlating variables are included. Total phosphorus was not included in this analysis as it significantly co-correlated with orthophosphate ($r = 0.98$, $p < 0.001$), and orthophosphate was selected because it is the most bioavailable form in rivers, and therefore most ecologically relevant to the microbial community.

Other variables that were not selected for this analysis include total nitrogen which was available for <50% samples (nitrite-N and nitrate-N were more complete and included), chloride which co-correlated with conductivity ($r = 0.84$, $p < 0.001$) and had fewer available measurements (403 vs 444 for conductivity), and ammonia-N which co-correlated with both conductivity ($r = 0.89$, $p < 0.001$) and ammoniacal-N ($r = 0.86$, $p < 0.001$). We have emphasised our selection choices in the Methods (lines 854-858), and we have now included all available environmental variables, including total phosphorus, in both the correlation analysis (Extended Data Fig. 9) and NMDS analysis (Extended Data Fig. 8, Supplementary Data 5) to improve interpretability of the patterns, and provided the raw data in Supplementary Data 1.

1.3. Moreover, since geology accounts for the largest proportion of variance, including metrics like CaCO₃ (i.e., alkalinity) would also be relevant—does alkalinity in rivers depend on surrounding geology? While the authors highlight geology as a strong driver (e.g., calcareous vs. siliceous substrates), it's unclear whether rivers were clustered or compared along those axes to support this conclusion more directly. Similarly, while geology and land cover are identified as primary drivers of biofilm communities, their influence likely occurs through changes in water chemistry. However, water chemistry explained little variation in community composition, raising the question of whether this relationship is adequately captured, or if other factors are more directly influencing biofilm composition. This could use more clarification and discussion.

We would like to emphasise that alkalinity was analysed in the original manuscript, and the dependence of alkalinity (and other water chemistry variables) on surrounding

geology and land cover were analysed and discussed (lines 352-355). However, we thank the reviewer for prompting us to explore this relationship further, and we have now included NMDS analysis to demonstrate clustering of samples along the geology gradient and the relationships with water chemistry (Extended Data Fig. 8). We have further elaborated on the relative influence of geology and water chemistry as suggested (lines 348-360), in addition to the influence of other factors that were not measured in the present study e.g., the biofilm matrix, which may buffer against conditions in the water column (lines 365-370).

1.4. Second, analyses are restricted to phylum-level, which may obscure ecologically meaningful variation at finer taxonomic levels. Incorporating genus-level trends could yield new insights, especially when paired with genome-resolved functional traits.

We thank the reviewers for their suggestion regarding genus-level trends. We want to reiterate that all the analysis was done at the metagenome-assembled genome (MAG) level, which are annotated down to the species resolution, as mentioned throughout the data analysis methods section. Clarifications have been made throughout to highlight this further (e.g., lines 832-834, 841-845, 846-849). Given the large set of MAGs recovered within this study, presenting and discussing the metabolic and variance partitioning information for all MAGs or even all genera individually would be challenging and lack clarity for readers. However, as suggested, we have now included figures presented at the genus level (Extended Data Fig. 5 and 7), highlighting the top 30 genera selected for based on their prevalence and abundance within the dataset, while MAG level gene abundances were provided in Supplementary Data 4 in the original manuscript.

1.5. Although the authors emphasize functional potential (e.g., genes for carbon and nitrogen utilization), variance partitioning shows only modest explanatory power for environmental variables. More discussion on microbe-microbe interactions within the biofilm matrix or other potential variables responsible for the presence of these genes is warranted.

We have discussed the relatively low explanatory power for certain environmental variables in more detail (lines 361-370) and further highlighted the role of microbe-microbe interactions within the biofilms and the possibility of this being another factor contributing to the variation observed (lines 402-406).

1.6. In short, the manuscript gestures at the importance of river biofilms and their functional capacity, but the analysis and interpretation stop short of delivering a clear conceptual advance. Many of the discussion points are overly broad and lack direct relevance to the results, which weakens their impact and clarity in relation to the study's findings. With stronger framing, more targeted use of the data, and a sharper interpretation of key findings, this study could make a more compelling contribution.

We thank the reviewers for prompting us to further the depth of our analysis and discussion. After making the suggested revisions, we believe the narrative, analysis and ecological advancements better represent the depth and complexity of the dataset. We also want to emphasise that this dataset is the first of its kind and serves as a critical baseline study for understanding the community assemblage and functions within river biofilms at a national scale.

Minor comments:

1.7. L1: The title should reflect the main findings of the paper. Currently, it reads too broad. Please refine.

We thank the reviewer for this comment and are happy to consider any suggestions that they and the Editor may have with respect to this.

1.8. L26: The sum total variance here is misleading since nutrients accounted for a smaller contribution compared to geology and land cover.

We have removed nutrients from the list to focus on only the most dominant environmental variables in the abstract and reworded for clarity (lines 27-29).

1.9. L31: This is mentioned in the provided citation, the relevance/importance of fungi in rivers systems should be noted, see <https://pmc.ncbi.nlm.nih.gov/articles/PMC4601154/pdf/emss-63879.pdf>

We thank the reviewers for highlighting this and have now updated the Introduction in line 36 to include all major phylogenetic clades, including fungi, that are relevant to biofilms, as reported in the paper by Besemer et al.

1.10. L35: Aquatic plants? Litter from riparian vegetation? Please clarify.

We have clarified that we are referring to aquatic plants on line 37.

1.11. L36: Microbe-microbe interactions (quorum sensing, cooperation/competition) also occur within the biofilm matrix and contribute to composition and function.

We highlighted that biofilms host interacting assemblages of microbes in this sentence, and as suggested, we have further elaborated on this important point in line 52-53. Further to this, we have also elaborated on interactions in line 62-63 following comments 1.13 and 1.14 below, highlighting interactions as a potential factor contributing to composition and function.

1.12. L41: How could these results help inform management? No mention of application other than it could, rather than how.

We have expanded the introduction with specific examples where understanding the diversity and functions may be applied to ecosystem management. See lines 42-47 in revised manuscript.

1.13. L43: Microbe-microbe interactions? Macroinvertebrate grazers?

See response to comments 1.11 (above) and 1.14 (below).

1.14. L47: Good to include a mention of biotic interactions.

As suggested, we have now highlighted biotic interactions (competition, cooperation, grazing) in our summary of factors influencing microbial community composition and function in lines 52-53 and lines 62-63.

1.15. L49: How is this limited? In what ways?

We have discussed the limitations to our knowledge of microbial biofilm diversity and function, many of which are the same as the limitations to identifying environmental drivers in comment 1.16 below, including the complex and heterogenous nature of

biofilms, the relatively small scale of previous studies (lines 59-68) and lack of metagenomic-based approaches (lines 76-78).

1.16. L51: What obstacles make this challenging? Moreover, <https://www.nature.com/articles/s41522-021-00251-2> discusses the advantage of forming biofilms in mitigating against environmental stressors as well as antagonistic microbe-microbe interactions—could biofilms be more resistant to change compared to water column, sediment or other sink locations in rivers?

Further to comment 1.15 above, we have updated the Introduction to highlight the specific obstacles that make identifying environmental drivers challenging, notably the multitude of environmental variables that may act on biofilms in different ways, the lack of studies that explore a full suite of environmental variables, and as suggested, the fact that the biofilms themselves could further complicate relationships due to buffering against fluctuations and complex microbe-microbe interactions (lines 61-64).

1.17. L71: Did you control for interannual variability? Seasonality likely plays a big role see <https://journals.asm.org/doi/10.1128/AEM.72.1.713-722.2006>
<https://www.nature.com/articles/s41598-020-73293-9>
<https://www.nature.com/articles/s41586-024-08240-z>
<https://journals.asm.org/doi/10.1128/aem.72.1.713-722.2006>

We agree that seasonality may have an effect. This analysis is part of a different hypothesis that is currently being tested for a submission in a future manuscript with increased power with respect to additional samples and sites for a more balanced design, owing to using a more cost-effective analysis (metabarcoding). We have therefore highlighted that seasonality may be a factor influencing biofilm community dynamics that requires further study on lines 391-395.

1.18. L72: Why were these sites chosen relative to others? Were the same sites sampled over the 3 year period?

These sites were selected by the Environment Agency (UK) using a randomised sampling design to create a network of sites with good spatial coverage of England, and capture all major land cover types, geologies and physicochemical conditions as explained and elaborated on in the revised manuscript in lines 86-92 and lines 692-694 in the Methods. Further to this, we have included a breakdown of how many samples were collected per year and how many sites were sampled in every year on lines 694-695.

1.19. L78: Do you have total Phosphorus? This is a limiting nutrient important for microbial metabolism.

Total phosphorus was measured but not included within the variance partitioning analysis for reasons explained in the response to comment 1.3 and lines 854-858 in the Methods. However, in response to the suggestions, we have now included total phosphorus in the correlation analysis (Extended Data Fig. 9, Supplementary Data 5) and NMDS analysis (Extended Data Fig. 8) to improve interpretation of the environmental relationships.

1.20. L85: This header should be tailored to the content of this section—primarily methods and results for GC association with env. variables. should be divided accordingly.

We have added a new section heading: 'Genomic traits of river biofilm bacteria' (line 149).

1.21. L88: Why? How is it different?

For reasons detailed in responses to comments 1.15 and 1.16, and there are few studies on the complexity of river biofilms, particularly at a large scale, compared to the others focused on the water column (for example, Borton et al. 2024, which we have also cited for clarity on lines 111-113).

1.22. L95: Can this be specified at a higher taxonomic resolution? Fungi, algae, etc. Algae are a large carbon source for heterotrophic microbes in biofilms see <https://www.sciencedirect.com/science/article/pii/S0923250815000819?via%3DiHub> The role of algae in biofilm function might explain some of the variation observed in this study.

For reasons outlined in line 120, bacterial diversity and function was the focus of this manuscript, and breaking down the proportion of reads assigned deeper than eukaryotes is therefore out of the scope of this manuscript. The eukaryotic contributions of biofilm communities are part of another ongoing manuscript with a targeted metabarcoding approach. However, we acknowledge the importance of eukaryotes and that interactions between domains could explain some variation, as we have now highlighted in response to comments 1.11, 1.13, and 1.14.

1.23. L100: It is surprising that on average only 2 distinct taxa per sample were found (~1000 taxa across ~400 samples). As sequencing depth was impressive at nearly 70M reads per sample, inadequate depth does not seem to explain this. The authors should discuss this in the context of their argument that these biofilms harbor mostly generalists. However, this also highlights whether dereplication was appropriate for this dataset. This step prevents the authors from detecting population level variance that may have corresponded more strongly with the measurement of environmental parameters.

We thank the reviewer for this comment. However, we must reiterate that the number of recovered MAGs does not represent 2 distinct taxa per sample. In fact, what we report in the study is the number of "non-redundant" taxa across the samples. As highlighted in line 122, the total number of MAGs recovered was 4,027 which were dereplicated using *drp* (line 792) to avoid overinflation of the functional gene detection and annotation, whilst mapping the reads per sample against the dereplicated set to obtain sample-specific abundances for taxa and genes. Since this study is focussed on the overview of the MAGs, their biogeography and their respective functions, the need for population-level variance is out of the scope. While we agree with the reviewer that this would be interesting, the analyses of environmental drivers at a population and pangenome-scale will be presented in future manuscripts focussing on key indicator taxa of the river ecosystem.

1.24. L108: Many microbial studies refrain from assigning bacterial taxa down to the species level. It is therefore unclear whether this finding that 95.1% of recovered MAGs represented previously uncharacterized species is truly a discovery of novel, undescribed taxa. This seems like it could have been caused by the choice of database or analysis pipeline.

As reported in the methods in lines 794-795, we used the GTDB database, which is a comprehensive collection of all the known and recorded archaeal and bacterial genomes, including reference genomes (<https://doi.org/10.1093/bioinformatics/btac672>). At the time of analysis, this collection comprised over 400,000 bacterial genomes and is curated. Therefore, we must respectfully disagree with the reviewer that the database was insufficient, reiterating our findings to be novel, or previously undescribed MAGs. Moreover, our analysis pipeline was built using the standard tools used for MAG identification, which includes comprehensive cutoffs for completion and contamination. To improve clarity of this aspect, we have updated the Methods in lines 794-785 to highlight the choice of database.

1.25. L111: Please highlight why Phylum-level was chosen as opposed to other taxonomic levels. Family- or order-level might yield new insights, especially in conjunction with predictive function see Shah et al. 2024 <https://enviromicro-journals.onLibrary.wiley.com/doi/10.1111/1462-2920.16634>

See response to comment 1.4. All our analysis were performed at the MAG level and subsequently grouped for visualisation at the phylum level to provide a more concise and clearer overview. We would like to highlight that all data and analysis is provided at the MAG level along with taxonomic assignment in the supplementary data files 2, 3, 4 and 5 in the original submission.

1.26. L117: What type of rivers? Similar rivers to those in this study, globally? Would also be good to mention if these rivers are rain or spring-fed since you also mention glacial-fed rivers.

We have added more detail and specified the type of rivers in the cited studies on lines 142-144.

1.27. L121: This statement relates to my concern about dereplication. Demonstrating that GC content or genome size varies across broad phylogenetic groups is not new. However, it is conceivable that further analysis of this dataset at finer taxonomic levels might reveal exciting patterns between environmental parameters and GC content and/or genome size. This paragraph as a whole seems problematic because of the coarse taxonomic scale used for this analysis.

As highlighted in our response to the comment 1.4 and 1.25, all the analyses were performed at MAG-level, while visual presentations are presented at the phylum-level for clarity. The relationship between GC and genome size is presented at the MAG level (Fig. 1E). In line with our response to 1.23 and 1.25, dereplication is necessitated for the scope of this study, where for the first time, we are reporting these genomic traits being driven by environmental variables in river biofilms.

1.28. L129: Is there a citation for this? If the citation is in the following sentence then please merge since it reads as a stand-alone.

We have added the citation to both sentences on lines 156-160 to be clear these are findings from the same study.

1.29. L130: Can you clarify the statement of stability? This reads as if nutrient-rich environments are less stable?

We have adjusted the wording of this statement on lines 159-160 to clarify that those with smaller genomes may be better suited to stable or oligotrophic environments.

1.30. L132: What hypothesis? That genome-size may confer an advantage in nutrient-specific environments? Please state more clearly.

We have revised this to directly state the hypothesis we are referring to on lines 161-164.

1.31. L136: Have you investigated other genomic parameters beyond gene size and %GC? see Shah et al. 2024 for work on ABC transporters and other adaptations across a large number of freshwater lakes <https://enviromicro-journals.onlinelibrary.wiley.com/doi/10.1111/1462-2920.16634>

Grazers have been found to influence biofilms, the idea of aquatic macroinverts as a source or dispersal mechanism of biofilms could be highlighted. Moreover, temperature, light, as well as surrounding riparian vegetation could also explain variation based on certain genes/metabolic pathways. If you have the data, this would be good to include. see Borton et al. <https://doi.org/10.1038/s41586-024-08240-z>

We thank the reviewer for this interesting comment. We have included several metadata variables in our analysis, including temperature, shade, other physicochemical parameters (see Supplementary Data 1 and 5), but we do not have data on riparian vegetation or macroinvertebrates, so are unable to perform that analysis.

We tested observations such as genome size and GC content, similar to the work by Shah *et al.* (Fig. 1E), and have now expanded on this as suggested with additional biofilm adaptations including ABC transporters, but also chemotaxis, flagellar assembly, EPS biosynthesis, quorum sensing, and the two-component system, see Extended Data Fig. 2 and lines 167-176.

1.32. L159: Would be good to bring in the Baas Becking hypothesis here for context.

We have related our findings to the Baas Becking hypothesis, as suggested, see lines 217-218.

1.33. L181: This finding suggests that dispersal is high within river systems, but are you seeing distinct compositions across rivers? Within rivers between upstream and downstream?

We tested the composition of the bacterial communities across rivers upstream vs downstream by exploring their relationship in the context of stream order but did not find significant associations (lines 392-394, Fig. 6E, Extended Data Fig. 9).

1.34. L189: What is known so far? The authors mention that "X is limited, or unknown" but do not provide much commentary or mention on the knowledge gaps themselves.

We have revised this and provide examples of the specific knowledge gaps, see lines 230-236.

1.35. L192: Semantics, but "contributed" implies activity but this study only addresses functional capacity. Descriptions should reflect this (e.g., "contribute to X based on genes encoding for X.").

We have reworded the sentence and others throughout this section to reflect this. See lines 237-261.

1.36. L194: A few more words are needed here to explain methodology. Where did these cycles come from? KEGG pathway database? Further, many of the findings in this paragraph are descriptive and do not differ from expectations based on previously described freshwater bacteria.

As suggested, we have added more detail to the methods outlining the METABOLIC tool used to identify metabolic pathways, which are identified using several databases integrated within the tool, including KEGG, see lines 798-800 and cited paper (<https://microbiomejournal.biomedcentral.com/articles/10.1186/s40168-021-01213-8>). Similar to this, we have also provided more detail outlining the metabolisHMM tool, see lines 800-802 and cited paper (<https://www.biorxiv.org/content/10.1101/2019.12.20.884627v2>). We reiterate that this study is the first of its kind, necessitating a broad overview of the various functions found in microbial biofilms in freshwaters. A study to this scale has not been undertaken, except for the one by Borton et al., who did not include biofilms in their analysis. We therefore respectfully suggest that a description of the identified functions is well within the scope of this manuscript.

1.37. L212: With no association with the environmental drivers of your study, how does this relate? This seems to be a list of all annotated genes without a framework of the broader relevance of these genes. See <https://www.nature.com/articles/s41522-021-00251-2> which also mentions that the biomatrix itself can serve as a resource for bacterial communities. It would be good to include more discussion on how dissolved biopolymers from the matrix could perhaps buffer potential effects from nutrient fluctuations or disturbance.

We thank the reviewer for this comment and have updated the discussion to further elaborate on how matrix biopolymers may serve as a nutrient resource and buffer the effects of nutrient fluctuations and other disturbances. See lines 283, 291-295.

1.38. L213: predicted functional capabilities?

As stated in our response to comment 1.35, we have reworded the section throughout to reflect these results are predicted capabilities or capacity based on identified genes lines 264-315.

1.39. L225: Penesyan et al 2020 <https://www.nature.com/articles/s41522-021-00251-2> explains that the stratification of river biofilms is dependent on location in the water column "shallow aquatic environments, microbial species are organised according to their metabolic and energetic properties" since you have depth data this could be useful to include. Moreover, could you clarify if this is related more to the env. variables of interest, or from the biofilm matrix?

The river depth data is an estimate of river depth from a physical characteristic map (lines 735-738) and was not measured in situ at the time of biofilm sampling. Although interesting, it would therefore not be appropriate to include the suggested analysis. We have clarified that we are referring to the microenvironment within the biofilm matrix itself on line 276.

1.40. L240: Cooperative and antagonistic interactions? Selective pressure of microbe-microbe interactions are also at play and should be mentioned. Did you notice community turnover across sampling years? How consistent are the communities over time?

In line with our responses to previous comments on including interactions as a factor, we have included the discussion of microbe-microbe interactions in lines 52-53, 62-63, 165-166 and 405-406. in the revised manuscript. We agree with the reviewer that annual turnover is an interesting question. However, as stated on line 695, only a small proportion of the sites were sampled in all three years and temporal analysis of the current dataset would therefore lack statistical power. This is out of the scope of the present manuscript and will be explored in a future manuscript when more temporal samples have been collected as part of routine monitoring.

1.41. L249: How precise are these genes for anthropogenic pollutants? It would be useful to clarify that these polycyclic aromatic compounds differ from naturally found aromatics, such as those comprising plant secondary metabolites.

This is a good point to highlight, and we have reworded the sentence to reflect that the identified genes are also relevant to naturally occurring compounds on line 305.

1.42. L245: Do the rivers in this study have a history of pollution? If so, from what type (mining, agriculture, sewage, etc.)?

The accurate historical data of pollution for the significant number of sites/ivers sampled across England that would be required to provide this information is not available, however, we do provide WWTP load, which is included within the variance partitioning analysis, discussed in lines 376-382, and provided in Supplementary Data 1.

1.43. L251: Do you have a table of these genes?

Yes, provided as Supplementary Data 4: metabolism_and_functions.xlsx along with the original submission.

1.44. L252: Similarly, can we be sure 'transport of metals' is related to industrial discharge rather than the transport of metals in natural systems without pollutants? (i.e. siderophores for Iron)

As above, we have reworded this paragraph to be clear the identified genes may be related to naturally occurring compounds/metals in addition to those from anthropogenic sources on line 308.

1.45. L266: Planktonic? Please clarify.

The cited studies all looked at benthic biofilms on stones, we have reworded this sentence for clarity on line 320.

1.46. L274: Please provide more detail for the other characteristics.

Details of the other characteristics are discussed in detail in the following paragraphs (WWTP load, line 376-382, stream order, line 392-393, river width and depth, line 393-395, and shading, line 395-396).

1.47. L275: Please state which factors.

We have reworded this to be clear that we are referring to the collective variance explained by all measured factors here on lines 331-332.

1.48. L281: Please identify which land cover types contributed most to composition.

We have identified the dominant land cover types as suggested on lines 340-342.

1.49. L291: Have you explored if these metrics differ between rivers grouped by geology type? The dataset also contains CaCO₃, so did you find if concentrations vary among your sites?

We have explored the relationships with water chemistry parameters, including alkalinity across the geology gradient further with NMDS analysis (Extended Data Fig. 8, lines 348-352).

1.50. L293: Perhaps a more precise phrase than ‘water chemistry’ is needed. The prior sentence discusses that pH and alkalinity do have important effects on biofilm communities. Water chemistry traditionally includes pH and alkalinity.

pH and alkalinity are included within the water chemistry category. We have reworded this section to avoid confusion, see lines 361-365.

1.51. L296: The Methods state that biofilms were scrapped from stones and macrophytes. Please rework this statement given the locations from which biofilms were sampled (not only stones)—it does not seem probable that microbes would directly rely on stone-derived nutrients. It would be interesting, however, to see if P water chemistry varies by geology since soil and bank erosion also contribute to P in river systems. See this paper on geologic N inputs <https://www.nature.com/articles/27410>

As explained in lines 695-698, biofilms were predominately sampled from stones with occasional exceptions from macrophytes, however, this information was not recorded due to the nature of the routine monitoring program (see response to comments 1.53 and 1.60). As explained in our response to comment 1.2, TP and TN were not analysed further due to data availability and co-correlation with other biologically relevant measures. However, we have removed the sentence (see response to comment 1.52 below), and we agree with the reviewers that this could be an interesting relationship, and as such have included TP in our NMDS analysis to show the relationship with geology (Extended Data Fig. 8).

1.52. L296: A bit of a stretch to include this unless you have chemistry data on the biofilm itself, otherwise should be stated with caution.

Considering we do not have chemistry data on the biofilm microenvironment itself, we have removed this statement as suggested and instead highlight that the relatively small variance explained by water chemistry may in part be due to the biofilm matrix buffering communities against fluctuations in the water (lines 365-370), as suggested by the reviewer in comment 1.37.

1.53. L320: Were there relevant methodological differences between the two studies? For example, it is unclear through this point in the manuscript whether the current study sampled biofilms of both organic and non-organic substrates.

Main methodological difference between the two studies (present study and Borton et al., 2024) which we highlight is surface water vs biofilm bacterial communities on line 391. See response to comment 1.51 and 1.60 – biofilms in the present study were predominately sampled from stones with occasional exceptions from macrophytes where stones were not accessible, however due to the large-scale routine monitoring program, it was not recorded for every site whether samples were stone or macrophyte biofilms.

1.54. L314: Do you find that cyanobacteria are more abundant in higher order streams? Or other photosynthetic Eukaryotes?

The focus of this manuscript is on bacteria, and eukaryotes are therefore beyond the scope of the research questions. However, we found that stream order accounted for <1% variation in any bacterial phylum, including cyanobacteria, as discussed in lines 392-394. Furthermore, cyanobacteria negatively correlated with stream order, indicating a lower abundance in higher order streams, although this correlation was not strong ($r=-0.16$, $p < 0.001$, Extended Data Fig. 9, Supplementary Data 5). We have discussed this in relation to other studies on biofilms, which also found similar relationships with stream order and the river continuum concept (RCC) in contrast to planktonic studies such as Borton et al. (lines 396-400).

1.55. Figure 1: Presents a detailed overview of MAG quality, taxonomic novelty, and phylogenetic diversity, which is useful for assessing the scope of the dataset. However, it functions more as a summary of dataset characteristics rather than a figure that advances the narrative or supports a central hypothesis. Consider moving this figure to the supplement and using main-text figures to focus more directly on ecological or functional insights derived from the MAGs.

We agree with the reviewers that figures illustrating the ecological or functional insights are important, and Figure 1 does provide a clear summary of dataset characteristics which is integral to the study and introduces the reader to the scope of the dataset. Furthermore, the phylogenetic diversity and taxonomic novelty in particular are key outcomes of the study which need to be highlighted in the main text. We have therefore decided to keep Figure 1 in the main text.

1.56. Figure 3: Legend needs to be more detailed.

We have added more detail in the legend about occupancy and niche breadth as requested, see lines 668-671.

1.57. Figure 4: Where did these pathways in a - c come from? These pathways seem to be overly simplified.

As outlined in our response to comment 1.36, we have now provided further detail to the METABOLIC tool which was used to generate these pathways using KEGG modules (see lines 798-800 and cited paper). In response to this comment, we have also included this information in the legend of Fig. 4, lines 672-677, and similarly for Fig. 5, lines 678-682.

1.58. Figure 5: This is a nice summary of biofilm function, but the major conclusions to be drawn from this are unclear.

We thank the reviewer for this comment and would like to highlight that the findings from microtrait (Fig. 5) are discussed in detail from line 264 to line 315 and Fig. 5 is cited throughout this section. Major conclusions are the prevalence of metabolic traits e.g., breakdown of organic compounds (lines 265-269 and lines 299-315), diversity of trophic capabilities (lines 270-280), and range of biofilm and stress-related genes (lines 281-298). Together these highlight the exceptional functional versatility of the biofilm bacteria (lines 418-424 in the conclusion). We have added a subheading for this section focusing on function and Fig. 5 for clarity and distinction (line 263).

1.59. L577: Was spatial sampling across and within years randomized? Certain biases could have arisen in the dataset if certain regions/landscapes were sampled in a single year whereas a different landscape was sampled in a different year. If randomization was not done, then showing limited year-to-year variation in important parameters would be useful (i.e. were any of these years significantly lower in flow rate/drought?).

Yes, the RSN was designed using an unbiased, randomised and spatially balanced approach to reduce pressure-based sampling (Brown et al., 2015 - <https://www.sciencedirect.com/science/article/pii/S1878029615003205>), and we have added more detail about the sampling design on lines 692-694.

1.60. L581: Were any statistical comparisons made between biofilms collected from river stone versus those collected from macrophytes?

This is not possible as samples were not recorded as being stone or macrophyte biofilms. The samples were collected as part of routine monitoring for the Water Framework Directive (WFD) assessment, and this information is not required for the trophic diatom index calculations that the monitoring program focusses on, and as such is not recorded. However, samples were predominantly stone scrapes, macrophyte scrapes were only performed at occasional sites where no stones were accessible as explained on lines 697-698 and comments 1.51 and 1.53.

1.61. L582: This looks like a commonly used mixture to preserve RNA, not DNA. This should be corrected to avoid confusion and a citation should be included to justify this mixture works well as an alternative to premade solutions (i.e. RNALater).

The buffer is an RNALater-like solution for nucleic acid preservation, and DNA extracted from biofilms stored in this buffer have been shown to be stable for >3 years (Warren et al., 2024). We have added this supporting reference to the methods on line 700.

1.62. L595: Measurements were taken during a three month period, which was prior to biofilm collection, but were these measurements taken exactly three months prior to sampling? If not, what range of timespans elapsed between nutrient measures and biofilm sampling? How well do these nutrient measurements correspond with the conditions that occurred at these sites during biofilm collection?

Measurements were taken throughout the 3-month period up to and including time of biofilm collection. This therefore included up to five independent measurements taken on separate occasions across a timespan of 3 months and averaged. This was done to capture longer term conditions, rather than being limited to potentially transient conditions at the time of sampling due to the variability of grab sampled water chemistry. We have expanded this to clarify on lines 714-716.

1.63. L649 and 658: Given this range of sample depths, median depth would be more informative than mean.

This is a good point, we have now included median raw (line 769) and filtered (778) reads per sample.

1.64. L689: These are well accepted standards, but should still be cited: <https://www.nature.com/articles/nbt.3893>

We have now included the citation for MAG standards on line 817.

1.65. L695: Please provide a more detailed (brief) explanation of this. Helpful for a broader audience See Finn et al 2020 <https://academic.oup.com/femsec/article/96/8/fiaa131/5863182>

We thank the reviewers' for prompting us to provide more information on Levins' niche breadth, which we have now expanded on in lines 832-840.

1.66. L704: Please identify the LMM adjustment used.

Maximum likelihood estimation was used, and we have amended the sentence on lines 847-848 to clarify.

Reviewer #3 (Remarks to the Author):

The study “National-scale biogeography and function of river and stream bacterial biofilm communities” by Thorpe et al. provide a more comprehensive understanding of the ecological dynamics of freshwater microbes and their drivers at a global scale. More specifically they highlight (i) the microbial community composition of the river biofilm across the 450 samples in 126 National River Surveillance Network sites, (ii) metabolic potential of recovered 1014 MAGs in C, N and S cycling were illustrated, and (iii) role of environmental drivers on variation in community composition. The concept of the study was quite good and findings are interesting.

However, out of the total MAGs obtained, authors included the low-quality MAGs with more than 10% contamination in metabolic analysis (line 195). Recent studies are generally considering the MAGs with less than 10% contamination for analysis (Borton et al. 2024. A functional microbiome catalogue crowdsourced from North American rivers. *Nature*; Garner et al. 2023. A genome catalogue of lake bacterial diversity and its drivers at continental scale. *Nature Microbiology*; Bowers et al. 2017. Minimum information about a single amplified genome (MISAG) and a metagenome-assembled genome (MIMAG) of bacteria and archaea. *Nature Biotechnology*; Singleton et al. 2024. *Microflora Danica: the atlas of Danish environmental microbiomes*. bioRxiv). I suggest to re-do the analysis without low-quality MAGs. It can misinterpret the results in terms of metabolic potential due to contamination. It is quite confusing as a reader in some places authors used only high-quality MAGs (line 109), in some places all MAGs were included. Along with this author did not discuss about the genes related to biofilm formations in MAGs as well the overall discuss was not well written. There are few more questions need to be addressed to improve the quality of the results and their interpretations.

We would like to thank Reviewer 3 for reviewing our manuscript, and for their helpful recommendations, which we have incorporated into our revised manuscript or clarified as necessary.

2.1. Authors need to remove the citation from the abstract.

Citations have been removed from the abstract.

2.2. Line 20: Are these 1014 MAGs high quality? and classified up to species or strain level

The 1,014 MAGs encompassed all dereplicated MAGs, however, in response to the reviewer’s suggestions above we have filtered out low-quality MAGs to ensure robust analysis and accurate annotation (lines 817-819). We now report results based on 820 medium and high-quality MAGs (see line 129). The 820 MAGs are classified to the species level where possible using GTDB (lines 794-795 and lines 130-131), and the proportion identified as novel genera and species remains robust with only minor changes to the percentage compared to when low-quality MAGs were included (i.e., 21.6 to 20.6% novel genera). We have amended our results, figures, and discussion accordingly throughout, and we would like to highlight that this quality filtering has not significantly changed any of our findings or conclusions.

2.3. Line 25: Environmental drivers, most notably geology, land cover, and nutrients, explained up to 90% of the variation in community composition. The value is quite high

than previously reported average variation. Authors need to clarify this? Is it for overall community composition or for individual taxa?

We originally reported total variance explained by the full suite of analysed environmental factors at the phylum level (highest total explained by phylum – ‘up to 90%’). However, for clarity and improved resolution, we have now reported the variance explained by all analysed environmental factors as an average across individual MAGs, see line 27-29.

2.4. There is no result and discussion heading in the Manuscript. Please add to separate the section from the introduction.

We have added this section heading line 108.

2.5. Line 78: Unit of measurement is not provided in “Extended Data Table 2”. Summary of water chemistry data.

Thank you for pointing this out, we have now included units for each measurement in the table legend.

2.6. Line 671: Which tools authors had used for dereplication of bins?

We thank the reviewer for highlighting this, and we have now added that we used dRep on line 792.

2.7. In the section “Metabolic and functional potential of river biofilm communities”, genes related to biofilm are not described. Freshwater system can be a source of antimicrobial resistance genes, it would be great if authors can find the MAGs carrying these genes and how their abundance vary across the river network. Authors could have described more about the bioremediation potential of these microbes.

Genes related to biofilms e.g., biofilm formation, are discussed on lines 281-295. We agree with the reviewer that reporting AMR genes would be an interesting application. However, this is beyond the scope of the present manuscript and would require in-depth analysis and therefore a separate manuscript which is currently in prep. Similarly, the bioremediation potential of these MAGs is currently being analysed for a future manuscript with in-depth analysis associated with particular phylogenetic groups

(<https://www.frontiersin.org/journals/microbiology/articles/10.3389/fmicb.2021.764058/full>). However, we have explored other biofilm related genes as suggested, including ABC transporters, chemotaxis, flagellar assembly, EPS biosynthesis, quorum sensing, and the two-component system in Extended Data Fig. 2, lines 284-285 and 167-176.

2.8. Freshwater is considered as one of the source of GHG emission, therefore author can find out the genomic potential of these MAGs in GHG attenuation and how these biofilms can play an important role in GHG mitigation, if possible.

The metabolic pathways identified using the METABOLIC tool are presented in Fig. 4, this includes pathways and genes which are associated with greenhouse gases, which we have now emphasised (lines 258-261).

2.9. Authors can also perform the microbial source tracking of MAGs across the riverine network to understand the microbial movement from source to the sink and which are the seeding microbial members of the biofilm, if possible

We acknowledge that this is an interesting point, however we do not have the required source and sink samples to perform source tracking.

2.10. Authors can perform differential abundant analysis for MAGs across the different sampling sites.

Due to the national scale survey nature of our dataset, differential abundance testing would not be appropriate as we do not have two discrete groupings.

2.11. Authors need to improve the discussion in the whole manuscript as it is not up to the standard of the journal.

We thank the reviewer for this comment and as suggested, following the many revisions made throughout, we have improved the Discussion to improve clarity and scientific reasoning.

Response to Reviewers

Manuscript ID: NCOMMS-25-17856A

Title: National-scale biogeography and function of river and stream bacterial biofilm communities

Reviewer #1 (Remarks to the Author):

Thorpe et al have made a nice effort to address many of our minor concerns from the prior round of review. Unfortunately, we feel that several of our major concerns have not been adequately addressed by this revision. Nevertheless, we appreciate that the geographic scope of this dataset will without question provide excellent baseline data for many future studies. However, we think that many of the findings are overstated and unsupported, often missing key comparisons to controls and benchmarks, and so this manuscript would still require significant revision prior to consideration for publication.

A thorough description is provided of the biofilm community and genes within these taxa. However, with no quantitative comparisons of these results to the microbes inhabiting the water column, or other biofilm communities inhabiting other environments, there is no context for evaluating what is unique about these biofilm communities in terms of taxonomy, genomic characteristics or functional potential. Thus, the majority of results reported in this manuscript are descriptive and fall short of advancing understanding of the field of microbial ecology. For example, a sizable portion of the manuscript is dedicated to comparing GC content and genome size of different bacterial phyla. However, it is well appreciated in the field that such traits vary across the tree of life and the authors have not directly responded to this concern. How different are these features in their samples vs. those from a broad range of samples? Further, the authors argue that given the presence of many genes within the C, N and S cycling pathways and that these communities are vital to the cycling of these nutrients in rivers. We may be misinterpreting their statements, but in essence our understanding is that they are arguing that the presence of C, N, and S cycling genes is a surprising finding, when of course many microbes sequenced from many sources would have genes involved in C/N/S cycling given the fundamental importance of these genes to life. How does the occurrence of genes within these pathways differ from any other complex environmental community of bacteria? Are genes in these cycles over-represented in riverine biofilms compared to the null expectation (such as in contrast to the occurrence of these genes in surface water communities)? These controls are essential in order to make any substantive conclusion about the role of riverine biofilms in C/N/S cycling, and the authors should be able to make these comparisons using publicly available data sets. The same holds for claiming the value of riverine biofilms in metal transport, as iron transport, for example, is essential for most life.

The manuscript makes large claims about the critical role of riverine biofilms in ecosystem function and “modulating fluxes of greenhouse gases in river ecosystems” (L261). It is not possible to conclude from metagenomic data alone that these communities have a sizable effect on greenhouse gas fluxes. Doing so requires in-situ

measurements or mesocosm studies in which water quality changes are measured between treatments with and without biofilm communities, or contrasting measures when biofilm communities vary in taxonomic composition. At a minimum, measures of gene expression are needed.

Overall, we think this manuscript is still of high potential interest, because the results presented in Figure 6 are very interesting. Our suggestion was to focus on these results and perhaps pare down the other data presented unless these key control analyses are added. A stronger manuscript could be developed that focuses specifically on these results of how various environmental parameters predict biofilm composition and function. This is an outstanding dataset, but a sizable amount more needs to be done to elevate the interpretability and impact of these results.

We would like to thank the reviewers for taking the time to review our revised manuscript, and we appreciate their recognition of the geographic scope and value of our dataset in “*providing excellent baseline data for many future studies,*” which reflects the overarching goal of our study.

We have carefully revised our manuscript to ensure that interpretations are balanced, precise, and fully supported by our data by adjusting our language throughout. For example, we have clarified our approach to species delineation (L131-136 and L829-831), highlighted that a prevalence of C, N and S cycling genes suggests a capacity to perform diverse biochemical roles which may include contribution to greenhouse gas fluxes in river ecosystems (L272-275), emphasised that genes such as metal transport genes are also involved in essential metabolism pathways (L325-327). These modifications improve the clarity and interpretability of our results and avoid implying causality beyond what our data support.

We appreciate the reviewer’s view that comparisons with biofilms or other sample types in different ecosystems could provide interesting insights. We have therefore incorporated several additional comparisons. For example, we have identified similarities and differences in the dominant phyla observed with other river biofilms and surface waters (L144-153), explored the relationship between genome size and GC content observed for specific taxa across biofilm, pelagic and sediment communities from other rivers and lakes (L160-164), highlighted that the prevalence of biofilm-associated (L182-186) and stress tolerance genes (L312-315) is conserved across biofilms from a range of freshwater environments, and more comprehensively addressed the key differences between our study and Borton et al. (2024), highlighting where further research is required (L409-427). These additions better place our findings in a broader ecological context.

However, we respectfully note that our study was not designed to test hypotheses regarding differences across environments or to identify features unique to river biofilms. The primary aims of this work, as stated in the introduction (L98-106), were to characterise the biogeography, functional potential and environmental drivers of river biofilm bacterial communities at a national scale. While we acknowledge that more extensive comparative analyses, such as “*quantitative comparisons of these results to the microbes inhabiting the water column,*” could be valuable, such analyses currently have substantial limitations due to methodological differences and the wide geographic spread of publicly available metagenomes. For example, the suggested additional analysis would require comparison with surface water metagenomes from North American rivers in GROWdb (Borton et al., 2024). However, as pointed out by

the reviewers' themselves in comment 12, there are confounding variables between the datasets, including methodological differences and the 5,000-9,000 km distance between the studies. It would therefore be difficult to determine whether observed differences reflect habitat type or spatial variation. We believe such a comparison would have limited interpretive value, and the suggested analysis would therefore require substantial additional data collection which is beyond the scope of this manuscript. We highlight this and the need for further comparative research utilising co-located biofilm and water column samples in our revised manuscript (L423-427).

Overall, these revisions have significantly strengthened the manuscript, clearly situating our work within the existing literature, maintaining alignment with the study's scope and aims, and establishing a national-scale genomic resource that will serve as an essential foundation for future comparative and experimental studies.

Our detailed responses to each individual comment are given below in blue text. Tracked changes have been made in the revised manuscript, and quoted line numbers correspond to the clean 'no mark-up' version of the revised manuscript.

Additional Specific Concerns

Title: there is no reference to a quite interesting aspect of this paper--- environmental parameters are predictive of biofilm biogeography

While we agree that this is an important aspect of our study, we believe our current title accurately reflects the central focus of the study which is the national scale biogeography and functional potential of river biofilm bacteria.

1. L121. Not following how a dataset with 311 genera has only 47 species?

We thank the reviewer for highlighting this. The statement in our manuscript, "...including 20 known phyla, 35 classes, 91 orders, 160 families, 311 genera and 46 species" (lines 130-131 in the revised version), refers to the MAGs that could be assigned a taxonomy at each rank, and how many of these taxonomies were unique phyla, classes, etc. Out of the 820 MAGs, 46 were assigned to species-level (all different species), and the remaining 774 did not have any species-level assignment, as discussed in the following sentence and presented in Fig. 1C. These 774 MAGs do not have representative genomes in the GTDB-Tk database and represent unknown or unnamed species. We also corrected a small typo in the manuscript, changing 47 to 46 species.

2. L133. Based on the information currently provided, we are still not fully convinced of their argument that 94.4% of recovered MAGs represented previously uncharacterized species. Could you provide mean ANIs between your MAGs and the closest hit to the GTDBK database? As bacterial 'species' are an area of continued debate, more clarity of exactly how species are being defined is needed.

Species were defined according to average nucleotide identify (ANI) with reference genomes in GTDB-Tk. GTDB-Tk follows the widely accepted standard that a genome is considered the same species as a reference genome if it shares >95% ANI across >65% of the aligned genome, as the developers have explained here: <https://github.com/Ecogenomics/GTDBTk/issues/190#issuecomment-542961623>.

Using these thresholds, 774 of the 820 MAGs (94.4%) lacked a suitable reference in the database and are therefore considered potential novel species. We have revised

lines 131-136 in the Results and Discussion and lines 829-831 in the Methods to clarify this, and the full GTDB-Tk output with ANI values is now provided in Supplementary Data 3.

3. L152. Lack of context. If one were to look at Myxococcota MAGs in a range of environments, would they all have a large genome with high GC content? Unclear if this is an advance in understanding about the Myxococcota in general, or those specific Myxococcota that happen to inhabit riverine biofilms.

We provided context in the original manuscript by noting that “A similar phylogenetic relationship between GC content and genome size has been observed in lake bacteria (Cabello-Yeves et al., 2019; Shah et al., 2024)”. However, we have now provided an additional reference further supporting this finding in river systems (Chiriatic et al., 2023) and included more detail to highlight that the patterns observed in genome size and GC content for Myxococcota, Pseudomonadota, Cyanobacteriota, and Bacteroidota specifically are consistent with those found in sediment and pelagic samples from lakes and other rivers in lines 160-164.

4. L160. Would be useful to add reviews about genome streamlining theory (Giovannoni et al. 2014).

We have included the suggested review about genome streamlining theory in line 168.

5. L167-176. The manuscript is strengthened by this addition of genes involved in biofilm formation but again lacks sufficient context. Are these same genes universally found in other biofilms? In other biofilms, has it been found that “these traits were distributed across MAGs spanning diverse taxonomic groups?” Or is this a unique finding to riverine biofilms?

We agree that additional context is useful, and genes associated with biofilm formation have been reported across diverse taxonomic groups in other freshwater biofilms. We have revised the text in lines 182-186 to reflect this and that our findings extend this observation to river biofilms.

6. L210. The extra information regarding niche breadth is appreciated, but I am concerned this may be circular logic. If certain taxa are found in nearly all samples (i.e. high occupancy) in a sample set collected across a wide range of habitat types, would these taxa not by definition need to tolerate a wide range of habitat types found in the sample set and thereby be considered generalists? More discussion on how to interpret these results would be helpful.

We thank the reviewer for this thoughtful comment. As the reviewer highlights, occupancy and niche breadth are related, as taxa occurring in many samples are more likely to be generalists. However, the two metrics are not identical. Occupancy measures the frequency of occurrence across samples, while niche breadth complements this by quantifying the evenness of relative abundance across samples. Taxa that are both widespread and evenly distributed would have a high occupancy and high niche breadth and therefore be classified as generalists. In contrast, taxa that are widespread but concentrated in particular habitat types would show high occupancy but low niche breadth and would therefore be classified as specialists. This approach allowed us to move beyond occupancy alone as a proxy and to provide a more mechanistic measure of ecological strategies. We have clarified this distinction in lines 219-221.

7. L214. While a weak relationship between community dissimilarity and geographic distance could indicate high dispersal, is this the only possibility? If dispersal rates were low but evolutionary change was also low, or pressures on community composition/species sorting were uniform across space, would we not also see a weak relationship?

We agree with the reviewer that this is important to highlight, and we have added the other possibilities in lines 228-230.

8. L277-280. These inferences are not supported by results in this study. There is no microspatial component to the sampling of these biofilms, at least as reported. So, it seems that the authors assume their diversity measures are a consequence of stratification. Authors should reduce causal language and remove statements that are not drawn from the presented results. Making such inferences could be acceptable if it is made clear that future studies are needed to prove causality.

The sentence was intended to provide context for the range of trophic capabilities observed, highlighting that biofilms are known from previous studies (Penesyan et al., 2021) to have a stratified structure. We have reworded this to remove causal interpretation as suggested, emphasising this as ecological context in lines 289-294.

9. L296-298. It is challenging to see how these claims are supported with no reference samples to compare to. Are these genes associated with stress tolerance strategies found in all bacteria? Are these genes significantly more prevalent in riverine biofilms, than for example the surface water communities?

We thank the reviewer for this thoughtful comment. Our aim was not to explore enrichment relative to other environments, but to highlight that a broad set of stress related genes were present in the biofilm MAGs. However, we have compared our results with other environments in lines 312-315, highlighting that stress tolerance genes are widespread in biofilm bacteria across different freshwater environments.

10. L310-311. Certain metals, such as Iron, are essential for cellular growth. As a consequence, most bacteria have means of transporting such metals, such as siderophores. To make a claim that this high prevalence of metal transporters is due to transporting and transforming harmful pollutants, the authors must provide adequate context. Is this “99.39% of MAGs” significantly greater than the proportion of MAGs found in other systems to have these same genes involved in transport of metals?

We do not make any claims as indicated by the reviewer that the observed “high prevalence of metal transporters is due to transporting and transforming harmful pollutants”. Rather, the presence of these genes suggests the possible capability to transport metal pollutants which may include natural or anthropogenic sources. However, we have now modified the statement to include that these genes are also widespread in bacteria for acquiring essential metals, in addition to their possible contribution to metal cycling in rivers in lines 325-327.

11. L382. This statement needs to be moderated. This study does not measure how biofilms with certain community compositions versus others change nutrient cycling or impacts water quality.

This sentence has been modified as suggested in lines 399-402.

12. L399-400. Comparison to the North American dataset is certainly warranted, but caution is needed when drawing interpretations between these two studies because there are multiple variables that differ. For example, the NA study includes much larger rivers, which could have driven the results in this study that stream order was predictive. So, we do not think it is a valid argument that stream order shapes surface water bacterial communities but does not shape riverine biofilm communities. There are confounding variables between the two studies that would need to be addressed before such a claim could be made.

We would like to emphasise that the finding that the RCC is not a strong driver of bacterial community composition in river biofilms is supported by Besemer et al. (2013) (lines 416-418). Although the difference in the stream order range between our study and the North American study was highlighted, we agree with the reviewer that this requires more discussion. We have therefore expanded on the differences between the studies in lines 423-425. We also highlight that further study, particularly utilising co-located surface water and biofilm samples, is required to fully understand the influence of the RCC on freshwater bacteria (425-237).

13. L418-421. It is difficult for a reader to appreciate how this is a 'diverse array of genes' and functions without any reference point. How does this compare to complex environmental bacterial communities in surface waters? Or other biofilms sampled in other environments? Again, here is where a contrast to bacterial communities in the surface waters could go a long way towards demonstrating why these riverine biofilms are potentially unique in their contributions to riverine function.

In this concluding paragraph, we highlight the diverse range of metabolic functions and genes detected within our national river biofilm dataset. Our study focuses specifically on characterising river biofilms, and direct comparisons to surface waters or other environments have limited value due to the reasons outlined by the reviewers in comment 12 and are beyond the scope of this work.

14. L433-435. Seems like an overreach. No measurements were taken for how biofilms alter water quality or public health. The link to biodiversity conservation is unclear. Making such claims necessitates functional measurements of how biofilms alter water quality.

We have revised the text in lines 457-460 to clarify that our findings provide baseline data that may support monitoring and management, rather than being interpreted as implying direct effects. We removed the less clear link to biodiversity conservation and public health to avoid overinterpretation.

Reviewer #2 (Remarks to the Author):

Reviewer #3 (Remarks to the Author):

The authors have comprehensively responded to all the comments, resulting in substantial improvements to the manuscript.

We would like to thank the reviewer for reading our revised manuscript and for their suggestions in the first round of peer review which helped substantially improve the quality and contribution of our work.